# A Unified Approach to Interpreting Knowledge Distillation for Large Language Models via Interactions

**Qingzhuo Wang** [* 1]  **Ruiyang Qin** [* 1]  **Zhenxin Qin** [1]  **Wen Shen** [1 †]  **Zhihua Wei** [1 †]

## Abstract

Despite the success of knowledge distillation (KD) in Large Language Models (LLMs), the underlying mechanism behind its efficacy remains unclear. In this paper, we propose a unified approach to explore the common mechanism of various KD methods using interactions. Specifically, we decompose the output score of the LLM into the sum of numerous interactions. Each interaction represents a nonlinear relationship involving a set of input variables (*e.g.*, words). Based on the decomposed interactions, we discover that the common mechanism underlying various KD methods is the sparsification of interactions, *i.e.*, student models retain fewer interactions for inference while suppressing other interactions to zero effects. Furthermore, we discover that the performance variance across different KD methods arises from their capabilities in handling complex interactions. A KD method typically yields better performance if it enables the student model to achieve higher sparsity of complex interactions. Motivated by these insights, we propose a plug-and-play loss function called Complex Interaction Penalty (CIP) to explicitly enforce the sparsity of complex interactions during the distillation process. Extensive experiments demonstrate that integrating CIP consistently improves the performance of diverse KD methods on both in-domain and out-of-distribution benchmarks.

---

[*]Wang and Qin contributed equally to this work and agree that the order of their names may be exchanged as useful to highlight their contributions in individual professional pursuits. [1]School of Computer Science and Technology, Tongji University, Shanghai, China. [†]Correspondence to: Wen Shen <wenshen@tongji.edu.cn>, Zhihua Wei <zhihua_wei@tongji.edu.cn>.

*Proceedings of the 43rd International Conference on Machine Learning*, Seoul, South Korea. PMLR 306, 2026. Copyright 2026 by the author(s).

## 1. Introduction

Although LLMs excel in text generation tasks, their practical deployment is constrained by substantial computing expense and high-quality training data. Therefore, KD (Hinton et al., 2015) serves as an essential technique to compress LLMs into compact student models while retaining their capabilities. Despite the empirical success of various KD methods (Kim & Rush, 2016; Lin et al., 2020; Agarwal et al., 2024; Gu et al., 2024; Ko et al., 2024; 2025), the underlying mechanism of *why* KD works remains unclear. Current research primarily focuses on improving the performance of KD, yet lacks interpretability of the distillation process: is there a unified mechanism that explains why knowledge distillation consistently enhances model performance?

In this paper, we aim to explore the common mechanism behind the efficacy of various KD methods, thereby offering better guidance for the distillation process. Recent research (Chen et al., 2024; Zhou et al., 2024; Ren et al., 2025; 2024) has leveraged game-theoretic interactions to explain the internal logic of LLMs. Inspired by these studies, we employ interactions to provide a unified interpretation of the learning process across various KD methods.

Specifically, let $x$ denote an input sequence with $n$ input variables (*e.g.*, words), indexed by the set $N = \{1, \ldots, n\}$. An interaction represents a nonlinear relationship associated with a specific combination of input variables. Considering the sentence $x = $ *"I am a green hand means that I am a"*, the words *"green"* and *"hand"* have distinct semantic meanings; their co-occurrence in the set $S = \{green, hand\}$ triggers a special interaction effect $I_S$, which pushes the LLM's inference towards the ground truth answer *"beginner."* Li & Zhang (2023) rigorously proved that the scalar output $v(x)$ of an LLM is always equivalent to the output of an interaction-based logical model $\phi(x) = \sum_{S \subseteq N} I_S$. That is, $v(x) = \phi(x) = \sum_{S \subseteq N} I_S$. Thus, the inference logic of an LLM can be equivalently explained by a set of interactions.

Based on interactions, we conduct a comprehensive analysis of various KD methods and uncover a unified mechanism similar to *Occam's Razor* (Blumer et al., 1987): **the essence of distillation lies in the sparsification of interactions**. This means that after distillation, the student model retains

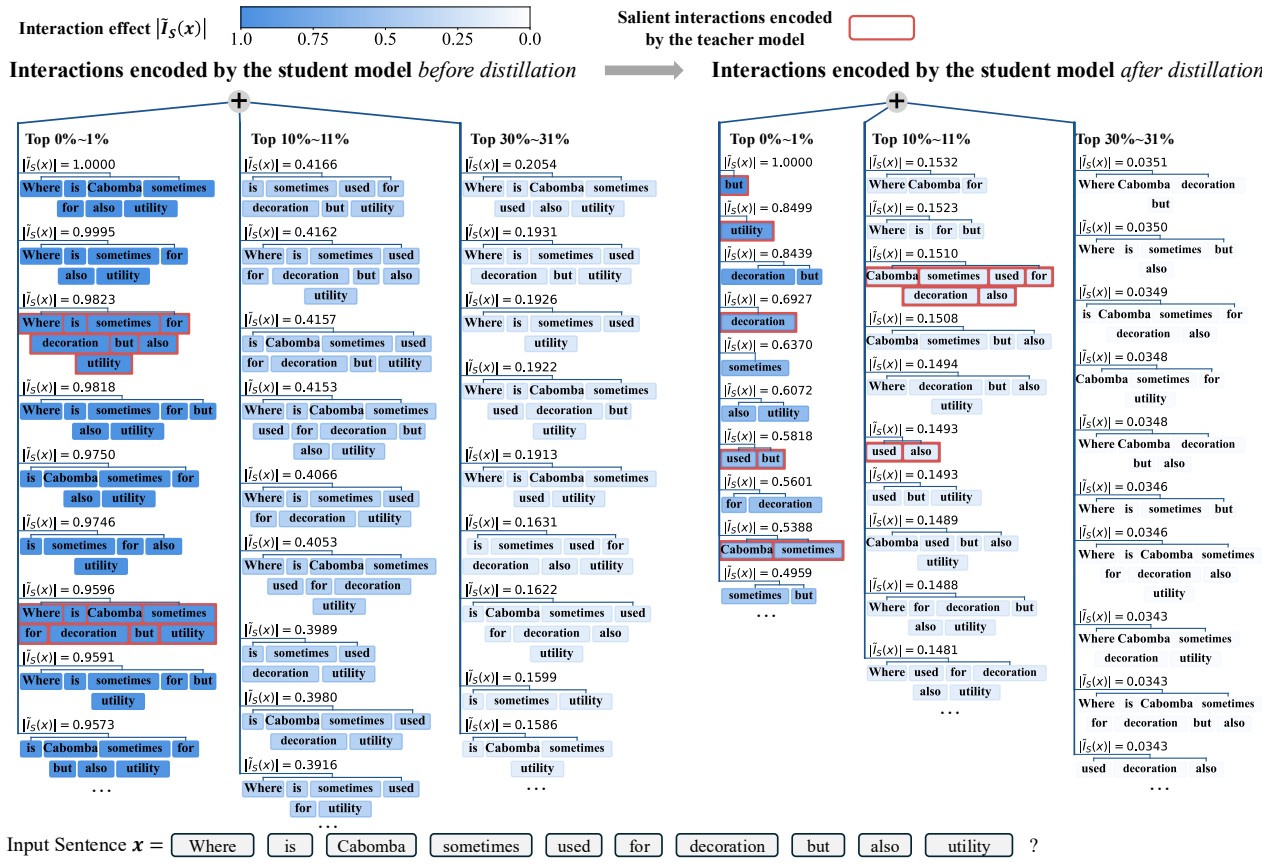

*Figure 1.* Comparing the interaction patterns encoded by the student model before and after distillation. The distillation process **drives the sparsification of interactions** by suppressing most of the interactions to near-zero effects, which is visualized by the fading color of the bars. Meanwhile, the student model **improves the alignment of interactions with the teacher model** by learning more salient interactions from the teacher model (highlighted by red boxes).

much fewer interactions with high effects (which we term *salient interactions*) and suppresses more interactions to near-zero effects than the original model. Crucially, we find that during this sparsification process, the student model does not retain random interaction patterns but rather retains salient interactions encoded by the teacher model. As shown in Figure 1, the student model after distillation retain more salient interactions learned from the teacher model than the model before distillation, while setting many other interactions to nearly zero effects.

Furthermore, we investigate the underlying reasons of the above mechanism from the perspective of interaction complexity. Specifically, the complexity of an interaction is defined by the number of input variables involved. A simple interaction indicates the relationship between few input variables, while a complex interaction represents the relationship between many input variables. By decomposing the overall sparsity of interactions into the sparsity of simple interactions and the sparsity of complex interactions, we identify that **the sparsification of complex interactions is the primary driver of the overall sparsification of in-**

**teractions**. In other words, a high proportion of complex interactions are suppressed, but a relatively small part of simple interactions are suppressed after KD. In addition, for salient interactions learned from the teacher model, a higher proportion of simple interactions are retained than complex interactions. We attribute this to the findings in Zhou et al. (2024): simple interactions typically encode generalizable knowledge essential for maintaining the model's general capabilities. In contrast, complex interactions are more likely to be considered as non-generalizable noises.

Based on the above insights, we explore why the performances of different KD methods vary. Empirically, we find that a KD method achieves better performance if it induces higher sparsity in complex interactions (*i.e.*, suppressing more complex interactions to nearly zero effects) and when it retains more salient complex interactions encoded by the teacher model. Successful KD methods act as a selective filter: they effectively suppress non-salient complex interactions which are mainly considered as noise by the teacher model while faithfully preserving salient complex interactions learned from the teacher model.

Based on the common mechanism we identified, we propose a plug-and-play loss term called **Complex Interaction Penalty (CIP)**. This penalty term explicitly enforces the sparsity of complex interactions during the training process. By integrating CIP into current KD methods, we demonstrate that it enhances the performance of various KD methods on both in-domain and out-of-distribution benchmarks.

## 2. Related Work

**Knowledge distillation.** KD (Hinton et al., 2015) and SeqKD (Kim & Rush, 2016) establish the foundation for model compression. To address the distribution mismatch, methods like ImitKD (Lin et al., 2020), GKD (Agarwal et al., 2024), and MiniLLM (Gu et al., 2024) utilize student-generated outputs for optimization. Recent advances focus on efficiency and performance bounds: DistiLLM (Ko et al., 2024) improves efficiency via Skew KL, DistiLLM-2 (Ko et al., 2025) integrates contrastive learning, and Speculative KD (Xu et al., 2024) bridges the teacher-student gap through interleaved sampling strategies (More KD methods are introduced in Appendix A). Regarding explanation for the mechanism of KD, existing studies fall into theoretical verification and representation analysis. Theoretical verifications (Phuong & Lampert, 2019; Allen-Zhu & Li, 2023) prove that KD accelerates convergence and transfers implicit "dark knowledge" inaccessible to single model. Representation analyses (Cheng et al., 2020; Ojha et al., 2023; Wu et al., 2024) observe that student models inherit conceptual features and causal mechanisms from teacher models. Furthermore, recent vision-based KD studies (Miles & Mikolajczyk, 2024; Miles et al., 2024) reveal that the student is forced to discard redundant noise and compress the teacher's knowledge into a sparser, more essential form during distillation. However, a unified framework is still missing to explicitly explain *the common mechanism of why various KD methods can improve model performance*.

**Using game-theoretic interactions to explain LLMs.** Conventional explanation methods (Tenney et al., 2020; Zhang et al., 2020b) suffer from a lack of mathematical faithfulness, implying their outputs may not accurately represent the internal logic encoded by the LLM. To bridge this gap, Ren et al. (2023a) proposed to use interactions between input variables to explain LLMs and provided a series of theoretical guarantees for the method's validity. Furthermore, Li & Zhang (2023) provided empirical evidence that interactions indeed represent meaningful concepts encoded by the LLM. Liu et al. (2023) discovered that simple interactions are more likely to be learned by LLMs than complex interactions. At the application level, the interaction framework has demonstrated its efficacy across a wide range of complicated tasks, such as adversarial transferability (Deng et al., 2024), model generalization (Chen et al., 2024; Zhou

et al., 2024), model training process (Ren et al., 2025; 2024), overfitting (Ren et al., 2023b) and other tasks (Shen et al., 2024; Li et al., 2025; Wen et al., 2026). In this paper, we use the interaction framework to analyze the process of knowledge distillation.

## 3. Preliminaries: Interactions

Given an LLM $v$ and an input sentence $x$ with $n$ words[1] indexed by $N = \{1, 2, \ldots, n\}$, let $v(x) \in \mathbb{R}$ denote the scalar output of the LLM. In practice, the scalar output $v(x)$ is defined as $v(x) = \log \frac{\bar{p}}{1-\bar{p}} \in \mathbb{R}$, where $\bar{p}$ denotes the probability of generating the ground-truth sequence in the text generation task. Specifically, given the ground truth sequence $y^* = (y_1^*, \ldots, y_L^*)$, we define $\bar{p}$ as the geometric mean of the probabilities of each ground truth token in $y^*$, *i.e.*, $\bar{p} = \left( \prod_{l=1}^{L} p(y_l^* | x, y_{<l}^*) \right)^{1/L}$. Please see Appendix C for why we choose this metric. Recent studies (Li & Zhang, 2023; Ren et al., 2023a) proved Theorem 3.1, which shows that the output score $v(x)$ on any randomly masked[2] input $x_T$ can be accurately calculated by the logical model $\phi(\cdot)$:

$$\phi(x_T) \triangleq \phi(x_\emptyset) + \sum_{S \subseteq N, S \neq \emptyset} \mathbb{1}(S \mid x_T) \cdot I_S, \qquad (1)$$

where $x_T$ denotes the input when we mask all the words in $N \setminus T$ and keep all the words in $T$ unchanged. The term $\mathbb{1}(S \mid x_T) \in \{0, 1\}$ serves as an indicator function for an **AND interaction pattern**, which represents an **AND relationship** between input variables in $S$. The scalar weight $I_S$ represents the **interaction effect**, quantifying the effect of an AND relationship. An AND relationship is activated only if every input variable within the set $S$ is present (unmasked) in the input $x_T$. Taking the sentence $x = $ *"I am a green hand means that I am a"* as example, the co-occurrence of the input variables in the set $S = \{green, hand\}$ pushes the surrogate logical model's inference towards generating the ground truth answer *"beginner"* by adding a numerical effect $I_S$ to it. If an AND interaction $S$ is activated, *i.e.*, $\mathbb{1}(S \mid x_T) = 1$, the interaction effect $I_S$ is added to the output of the logical model. Otherwise, if any word in $S$ is masked and the AND interaction is not activated, *i.e.*, $\mathbb{1}(S \mid x_T) = 0$, the interaction effect $I_S$ is not added to the output of the logical model.

**Theorem 3.1** (Universal matching property, proven in Appendix D.1). *Consider an input $x$ with $n$ input variables. Let $\{x_T \mid T \subseteq N\}$ denote the set of all $2^n$ different masked inputs. For every masked input $x_T$, when*

---

[1] We take words instead of tokens as input variables because different LLMs may divide the same word into different tokens. For example, GPT-2-0.1B tokenizes the word "Elements" into two tokens "Element" and "s", while LLaMA-7B treats it as one token.

[2] We employ standard masking technique, such as replacing the target token with a specific [MASK] token. Please see Appendix B for an introduction to masking strategies.

the scalar weight $I_S$ in the logical model $\phi(\cdot)$ are set to $I_S = \sum_{S' \subseteq S} (-1)^{|S|-|S'|} \cdot v(x_{S'}), \phi(x_\emptyset) = v(x_\emptyset)$, the output of the logical model $\phi(\cdot)$ can always match the LLM's output score $v(\cdot)$.

$$\forall T \subseteq N, \ v(x_T) = \phi(x_T) \tag{2}$$

Equation (1) and Theorem 3.1 establish a rigorous theoretical foundation, allowing us to treat interactions as the detailed inference patterns that constitute the model's internal logic.

# 4. Explaining KD via Interactions

## 4.1. Extracting Interactions in LLMs

In this paper, we use interactions as an analytical tool to investigate how interaction patterns of the student model change before and after distillation, thereby analyzing the underlying mechanism of the distillation process.

*Experiment setup.* We conduct experiments on six widely used KD methods, including KD (Hinton et al., 2015), SeqKD (Kim & Rush, 2016), ImitKD (Lin et al., 2020), MiniLLM (Gu et al., 2024), GKD (Agarwal et al., 2024), and DISTILLM (Ko et al., 2024). For comparison, we employ the original pre-trained student model without any further training as the **Base** model and the student model trained via standard supervised fine-tuning as the **SFT** model. We analyze three LLM families with the following teacher-student configurations: (1) GPT-2 family (Radford et al., 2019), we use GPT-2-1.5B as the teacher model and select GPT-2-0.1B, 0.3B, and 0.7B as student models; (2) OPT family (Zhang et al., 2022), we use OPT-13B as the teacher model and select OPT-1.3B, 2.7B, and 6.7B as student models; and (3) LLaMA family (Touvron et al., 2023), we use LLaMA-13B as the teacher model and select LLaMA-7B as the student model. All student models are distilled on the `databricks-dolly-15k` dataset (Conover et al., 2023). We follow Ko et al. (2024) to split the dataset into training, validation and test set. We evaluate model performance using the ROUGE-L metric (Lin, 2004) and compute interactions on the test set. For computational feasibility, we take words in the *"Instruction"* segment of the input as input variables. Meanwhile, the system prompt is treated as fixed, unmasked background context. Please see Appendix E for the details of implementation.

Figure 2 plots the probability density of absolute values of the normalized interaction effects for a representative input sample $x$. Specifically, the normalized interaction effect is calculated as $\widetilde{I}_S = \frac{I_S}{Max}$, where *Max* is the maximum absolute values of all $2^n - 1$ interaction effects of $x$. Results show that the non-distilled model exhibits a relatively flat and uniform distribution of interactions. In contrast, the distributions of distilled models exhibit an obvious peak near zero, indicating that the majority of interactions are suppressed to

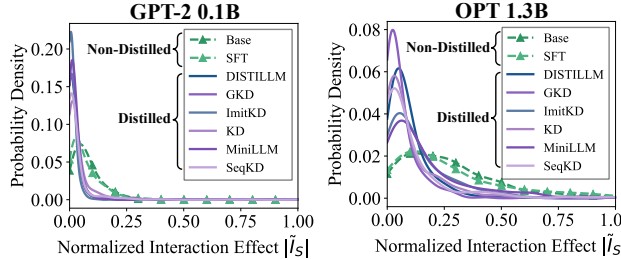

*Figure 2.* Comparison of the probability density between distilled and non-distilled models. Non-distilled models exhibit a flat distribution, while distilled models exhibit sharp peaks near zero.

nearly zero effects. It demonstrates that the distribution of interactions turns more sparse after the distillation process. Notably, we observe that SFT models do not exhibit similar level of sparsification as distilled models. We hypothesize that **sparsification of interactions is a unique mechanism of KD, distinguishing it from standard training methods**.

## 4.2. Interaction-Based Metrics for Analyzing KD

Inspired by the phenomenon, in this subsection, we propose three interaction-based metrics to analyze how a KD method change the interactions encoded by the student model.

**Interaction sparsity.** To evaluate the uniformity of the interaction distribution encoded by LLMs and thereby quantify the sparsity of interactions, we introduce the **Gini coefficient** and **Shannon entropy**.

*Gini coefficient.* Let $\mathcal{I} = \{|\widetilde{I}_S| : S \subseteq N, S \neq \emptyset\}$ denote the set of absolute value of all normalized interaction effects extracted from the input $x$. Let $M = |\mathcal{I}| = 2^n - 1$ denote the number of all the interactions. Let the values in $\mathcal{I}$ be sorted in ascending order as $u_1 \leq u_2 \leq \cdots \leq u_M$. For the input $x$, the Gini coefficient $G(x)$ is defined as:

$$G(x) = \frac{2\sum_{i=1}^M i u_i}{M \sum_{i=1}^M u_i} - \frac{M+1}{M}. \tag{3}$$

*Shannon entropy.* The Shannon entropy $H(x)$ (abbreviated as entropy) is defined as $H(x) = -\sum_S p_S \log p_S$, where $p_S = |\widetilde{I}_S| / \sum_{S'} |\widetilde{I}_{S'}|$ is the normalized probability. A high $G(x)$ or low entropy $H(x)$ indicates a sharp interaction distribution, reflecting high sparsity of interactions.

**Interaction alignment.** To evaluate whether the student model inherits the critical interaction patterns of the teacher model during distillation, we employ the overlap rate of interactions to measure the alignment of the salient interactions between the student and teacher model. We define a threshold ratio $k \in (0, 1]$. Let $\Omega_{\text{teacher}}^{(k)}$ and $\Omega_{\text{student}}^{(k)}$ represent the sets containing the indices of the top-$\lfloor k \times M \rfloor$ interactions with the largest absolute effects in the teacher and student models, respectively. For the input $x$, the student-teacher

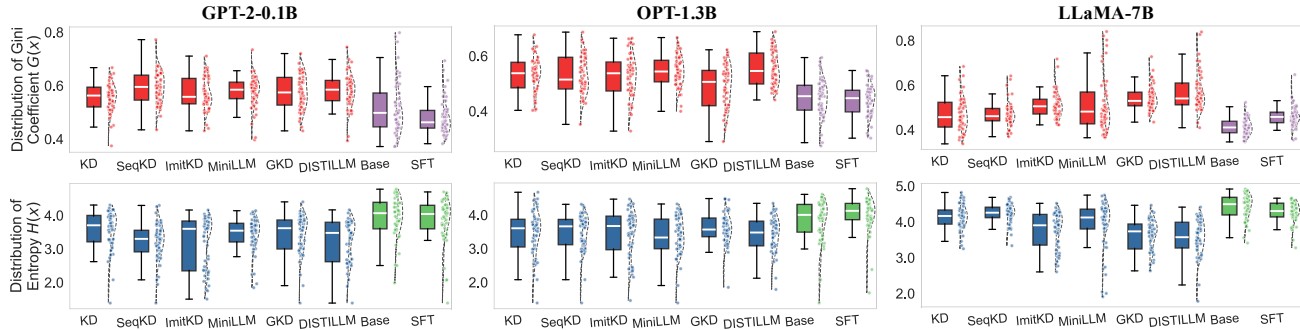

*Figure 3.* The distribution of the Gini coefficient and entropy across all samples. The results demonstrate that, compared to non-distilled models (Base models and SFT models), distilled models consistently exhibit higher Gini coefficients and lower entropy. It indicates that the interactions encoded by the LLM become more sparse after distillation.

*Table 1.* Comparing the student-teacher overlap rate of salient interactions before and after distillation. $\mathbb{E}_x\left[Overlap@k(x)\right]$ represents the average overlap rate across all samples.

| $\mathbb{E}_x\left[Overlap@k(x)\right]$ | Model | non-distilled | distilled (with different KD methods) | | | | | |
|---|---|---|---|---|---|---|---|---|
| | | Base | KD | SeqKD | ImitKD | MiniLLM | GKD | DISTILLM |
| | GPT-2-0.1B | 19.81% | 24.84% | 26.96% | 23.05% | 23.68% | 26.69% | 26.38% |
| $k = 0.1$ | OPT-1.3B | 21.96% | 25.24% | 26.48% | 21.47% | 26.54% | 25.23% | 25.71% |
| | LLaMA-7B | 21.35% | 24.18% | 21.55% | 22.02% | 27.62% | 29.43% | 31.60% |

overlap rate $Overlap@k(x)$ is defined as:

$$Overlap@k(x) = \frac{\left|\Omega_{\text{teacher}}^{(k)} \cap \Omega_{\text{student}}^{(k)}\right|}{\lfloor k \times M \rfloor}. \qquad (4)$$

A high overlap rate indicates that the student model has effectively captured the teacher model's salient interactions, implying a strong logic alignment.

### 4.3. Exploring the Common Mechanism of KD Methods

Based on the proposed metrics, we discover the following two common mechanisms shared by different KD methods.

**Distillation enhances interaction sparsity.** We employ the Gini coefficient $G(x)$ and Shannon entropy $H(x)$ to quantify interaction sparsity across the whole dataset. Figure 3 illustrates the distributions of these metrics across all samples. Results[3] show that distilled student models demonstrate an obvious distributional shift towards higher Gini values and lower entropy compared to corresponding base models and SFT models. It proves that the distillation process enhances the sparsity of interactions encoded by the student model.

**Distillation enhances interaction alignment.** The above phenomenon is counter-intuitive. Although KD suppresses most of the interactions to nearly zero effects and makes the student model rely on fewer salient interactions for inference, it conversely improves the performance of the student model. To explain such a counter-intuitive phenomenon,

---

[3]Results on more model family and model size in Appendix G.2 exhibit the same conclusion.

we further analyze the relationship between the salient interactions retained by the distilled student model and those encoded by the teacher model by the metric $Overlap@k(x)$.

Table 1 shows that distilled student models exhibit a higher overlap rate with the teacher model in terms of salient interactions than that of the non-distilled student models[3]. This indicates that KD drives student models to learn more salient interactions encoded by the teacher model, which accounts for the enhanced performance of the distilled model.

In conclusion, the sparsification of interactions demonstrates that the distillation process effectively compresses interaction patterns encoded by the student model. Furthermore, the alignment of salient interactions between the student model and the teacher model confirms that the compression is achieved by selectively **learning the teacher model's salient interactions** while **discarding other interactions**.

### 4.4. Exploring the Underlying Reasons behind the Common Mechanism

The mechanism of KD can be summarized as *"discarding the dross and selecting the essential."* To explore the underlying reason of the mechanism, we further identify which specific types of interactions are primarily treated as *"dross"* to be discarded, and which specific types of interactions are treated as *"essential"* to be selected.

To answer the above question, we analyze interactions with different complexities. The complexity of an interaction $S$ is defined as the number of input variables involved, *i.e.*, $|S|$.

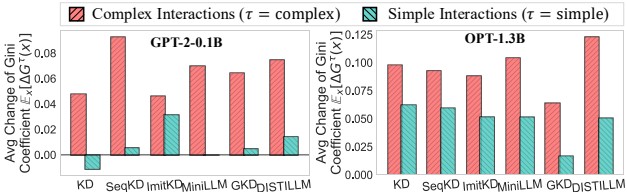

*Figure 4.* Comparison of the changes in Gini coefficients between distilled and base models for simple versus complex interactions.

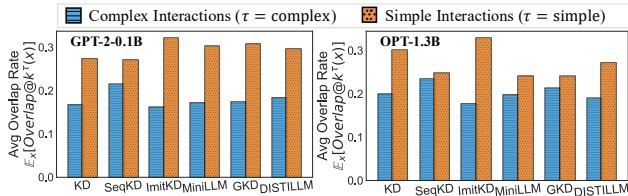

*Figure 5.* Comparison of student-teacher overlap rate between simple interactions and complex interactions after distillation. Results show that the student-teacher overlap rate of simple interactions is higher compared to complex interactions after distillation.

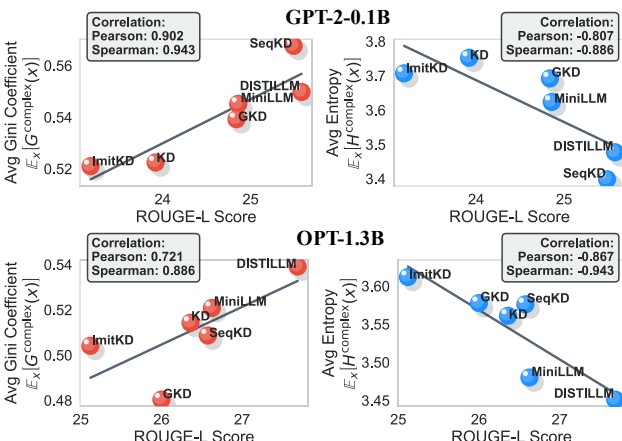

*Figure 6.* Correlation between the sparsity of complex interactions and model performance. Results show that increased sparsity (higher Gini coefficients and lower entropy) of complex interactions is positively associated with higher ROUGE-L scores. However, there is no distinct correlation between the sparsity of simple interactions and model performance, as shown in Figure 16.

Based on this, we partition interactions into two types: simple interactions, which represent relationships involving few input variables, and complex interactions, which represent relationships involving many input variables. In this way, the overall sparsity of all interactions can be decomposed into the sparsity of simple interactions and the sparsity of complex interactions. Following Li et al. (2025), we partition interactions into two groups: $\Omega^{\text{simple}}$ and $\Omega^{\text{complex}}$. Given an input sentence $x$ with $n$ words indexed by $N = \{1, \ldots, n\}$, we define $\Omega^{\text{simple}} = \{S \subseteq N \mid 1 \leq |S| < \beta\}, \Omega^{\text{complex}} = \{S \subseteq N \mid \beta \leq |S| \leq n\}$, where we set $\beta = \lceil n/3 \rceil$. To verify the robustness of the choice of $\beta$, we also conduct experiments on $\beta = \lceil n/5 \rceil$ and $\beta = \lceil n/2 \rceil$. Results in Appendix G.3 show that our findings are consistent when choosing different $\beta$. Accordingly, we denote the metrics in Section 4.2 for interaction type $\tau \in \{\text{simple}, \text{complex}\}$ and model type $\mathcal{F} \in \{\text{base}, \text{distilled}\}$ as $G_{\mathcal{F}}^{\tau}(x)$, $H_{\mathcal{F}}^{\tau}(x)$, and $\text{Overlap@}k_{\mathcal{F}}^{\tau}(x)$, respectively. $G_{\mathcal{F}}^{\tau}(x)$ is computed by sorting all interaction effects in ascending order in the subset $\Omega^{\tau}$. $H_{\mathcal{F}}^{\tau}(x)$ is derived from a probability distribution *re-normalized* within the subset $\Omega^{\tau}$, defined as $p_S = |\tilde{I}_S| / \sum_{S' \in \Omega^{\tau}} |\tilde{I}_{S'}|$. $\text{Overlap@}k_{\mathcal{F}}^{\tau}(x)$ is defined as the overlap rate of top-$k$ interactions ranked *exclusively* within the subset $\Omega^{\tau}$.

First, we aim to explore whether the sparsification of interactions is driven mainly by the sparsification of simple interactions or complex interactions. Figure 4 plots the change of Gini coefficients between distilled and base models for both simple and complex interactions, which is defined as $\Delta G^{\tau}(x) = G_{\text{distilled}}^{\tau}(x) - G_{\text{base}}^{\tau}(x)$. Results[3] show that the sparsity of complex interactions exhibits an obvious increase after distillation. In contrast, the sparsity of simple interactions exhibits lower increases or even decrease in some cases. This indicates that *the sparsification of complex in-*

*teractions contributes more to the overall sparsification of interactions after distillation.* In other words, the interactions discarded by the LLM during the distillation process are more likely to be complex interactions.

Second, we investigate whether the salient interactions learned from the teacher model are mainly simple or complex interactions. Specifically, we compare the overlap rate of simple interactions against that of complex interactions between the teacher and student models. Figure 5 shows that the overlap rate of salient simple interactions consistently surpasses that of complex interactions[3]. This indicates that *the student model learn more simple salient interactions than complex salient interactions from the teacher model*.

Thus, we can conclude that **complex interactions are primarily treated as *"dross"* to be discarded and simple interactions are primarily treated as *"essential"* to be selected during the distillation process**. We attribute this phenomenon to the findings of Zhou et al. (2024), which suggest that simple interactions typically encode fundamental, generalizable knowledge, while most of the complex interactions are considered as non-generalizable noise. Thus, the underlying reason of the common mechanism is that **the distillation process makes the student model retain more simple interactions to stabilize the student model's general capabilities, while discarding more complex interactions to filter out non-generalizable noise**.

### 4.5. Explaining the Performance Variance across Different KD methods

We further explore *why the performances of different KD methods differ* from the perspective of interactions. Specif-

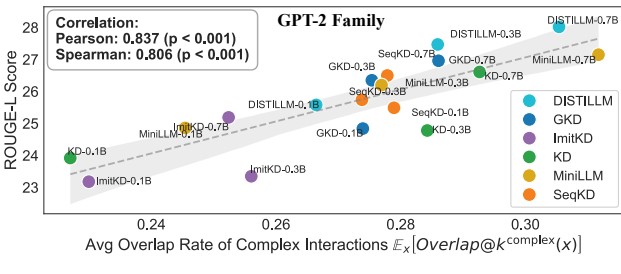

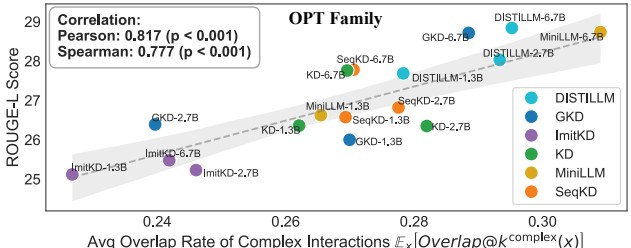

*Figure 7.* Correlation between the student-teacher overlap rate of complex interactions and model performance. Results show that increased student-teacher overlap rate of complex interactions is positively associated with higher ROUGE-L scores.

ically, we discover the correlation between the sparsity of complex interactions and the model performance of student models across different KD methods. As shown in Figure 6, **the sparsity of complex interactions is positively correlated with model performance**[3]. Student models that exhibit higher Gini coefficients and lower entropy (indicating higher sparsity) in their complex interactions generally achieve superior ROUGE-L scores. Furthermore, we analyze the relationship between the student-teacher overlap rate of complex interactions and model performance. As shown in Figure 7, the results[3] reveal **a positive correlation between model performance and the overlap rate of complex interactions**. This indicates that student models with higher performance learn more complex interactions from the teacher model compared to student models with lower performance. Counter-intuitively, the overlap rate of simple interactions shows no distinct correlation with model performance, as shown in Figure 18 in Appendix G.2. Across all the results, the Pearson correlation coefficient between simple interactions and model performance is $0.34$ ($\pm 0.12$), whereas for complex interactions, it is $0.81$ ($\pm 0.11$). This implies that while retaining more simple interactions is a common reason for general performance improvement in all KD methods, it does not account for the performance gap between different KD methods. Instead, the retainment of complex interactions serves as the decisive factor.

In conclusion, the performance variance across KD methods is related to their abilities in handling complex interactions. In general, superior performance is achieved when student models exhibit higher sparsity and higher alignment (overlap rate) with the teacher model in complex interactions.

## 5. Guiding KD via Interactions

In Section 4.5, we find that the sparsity of complex interactions is positively correlated with model performance. The more complex interactions are suppressed to nearly zero effects, the better the model performance becomes. Based on this insight, we introduce a plug-and-play loss function called **C**omplex **I**nteraction **P**enalty (**CIP**), which is designed to suppress complex interactions encoded by the student model during distillation. We define the loss function $\mathcal{L}_{\text{CIP}}$ as follows.

$$\mathcal{L}_{\text{CIP}} = \mathbb{E}_x \left[ \mathbb{E}_{S \in \Omega^{\text{complex}}} \left[ |I(S)| \right] \right] \tag{5}$$

According to Equation (5), directly computing all complex interactions is NP-hard. To alleviate computational complexity while preserving efficacy, we adopt a sampling method to approximate $\mathcal{L}_{\text{CIP}}$ by following Zhang et al. (2020a) and Li et al. (2025). Specifically, given an input sequence $x$ with $n$ words indexed by $N = \{1, \ldots, n\}$, we randomly partition the set $N$ into $m$ disjoint subsets $S_1, S_2, \ldots, S_m$, such that $\bigcup_{i=1}^{m} S_i = N$ and $S_i \cap S_j = \emptyset$. Based on this, we define the approximated loss term $\mathcal{L}'_{\text{CIP}}$ as follows.

$$\mathcal{L}'_{\text{CIP}} = \mathbb{E}_x \left[ \mathbb{E}_{K \subseteq \{1,\ldots,m\}, K \neq \emptyset} \left[ \left| I \left( \bigcup_{i \in K} S_i \right) \right| \right] \right] \tag{6}$$

To ensure that any sampled interaction $\bigcup_{i \in K} S_i$ is a complex interaction, that is, $|\bigcup_{i \in K} S_i| \geq \lceil n/3 \rceil$, $m$ must meet the requirement $m \leq \frac{n}{\lceil n/3 \rceil} \leq 3$. The detailed proof is in Appendix D.2. Therefore, the approximation method effectively reduces the complexity from $2^n$ to $2^m$, making the computational cost negligible compared to other steps in the distillation. See Appendix G.1 for the comparison of training time with and without the CIP loss.

The total loss function for KD is defined as the weighted sum of the original distillation loss $\mathcal{L}_{\text{KD}}$ and the CIP loss:

$$\mathcal{L} = \mathcal{L}_{\text{KD}} + \lambda \mathcal{L}'_{\text{CIP}}, \tag{7}$$

where $\lambda$ is the weight of the CIP loss ($\lambda > 0$).

We integrate the proposed CIP loss into six widely used KD methods and train on three LLM families, same as the experimental setup in Section 4.2. The LLMs are distilled on the `databricks-dolly-15k` dataset and evaluated across four diverse instruction-following benchmarks, including DollyEval, SelfInst (Wang et al., 2023), Super-Natural Instructions (Wang et al., 2022), and Vicuna evaluation (Chiang et al., 2023), to comprehensively assess their capabilities. We adopt two metrics for evaluation: ROUGE-L (Lin, 2004) and GPT-5 feedback (Zheng et al., 2023). See Appendix F for details on the experimental setup.

Figure 8 presents a detailed performance comparison across varying models and KD baselines, both with and without the CIP loss function (Here the weight $\lambda$ is set as 0.001 for

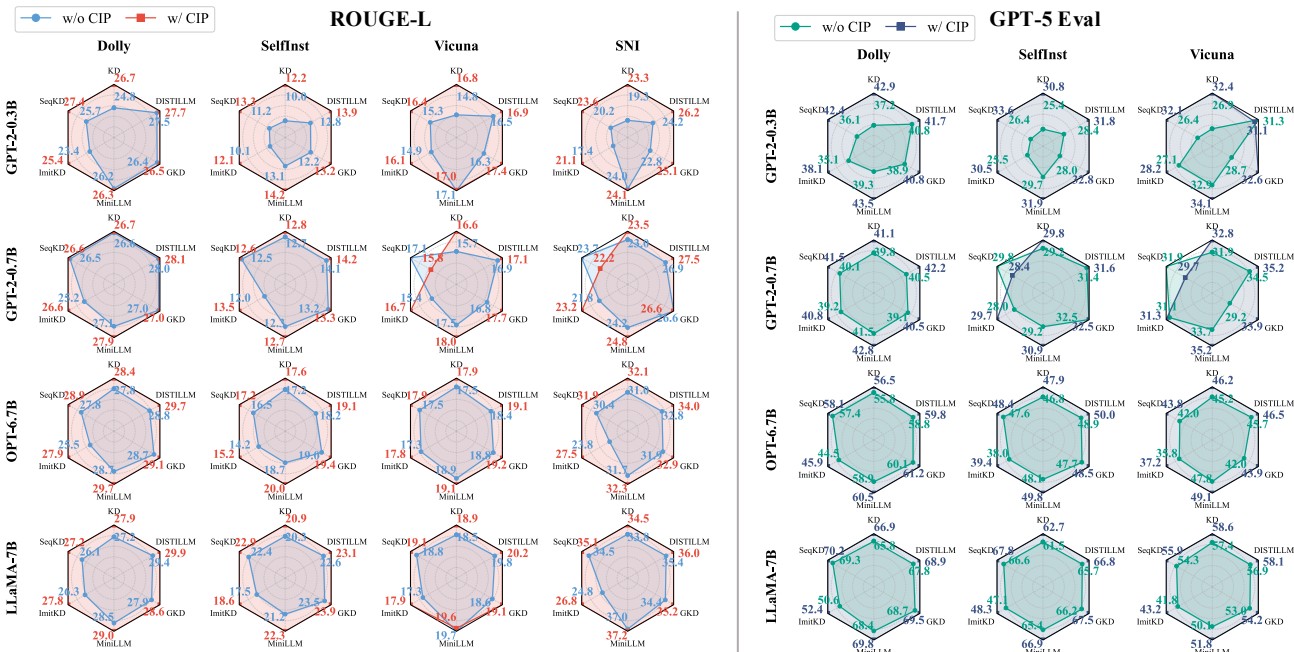

*Figure 8.* The comparison of ROUGE-L and GPT-5 scores between different KD methods without and with the CIP loss function across various model families and model sizes. Results show that adding the CIP loss improves the model performance on almost all the KD methods and datasets. Please see Table 6 in Appendix G for results of all models with the CIP loss.

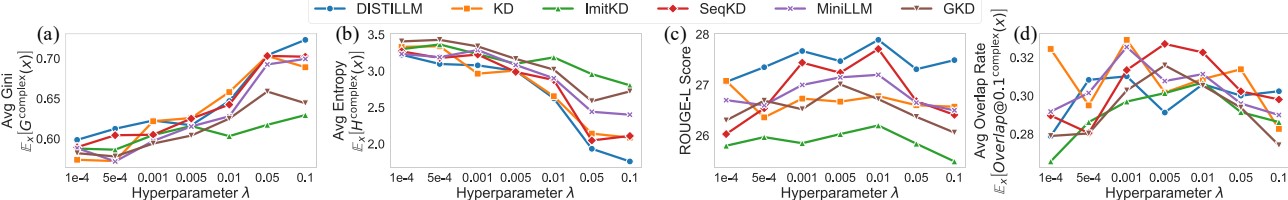

*Figure 9.* Exploring the effects of the weight $\lambda$ for on the sparsity of complex interactions (a and b), the performance of the student model (c), and the student-teacher overlap rate of complex interactions (d). Here the base model is GPT-2-0.3B.

all the models). Results show that **adding the CIP loss yields improvements in ROUGE-L and GPT-5 scores in most of the settings, regardless of the specific model or KD method**. Crucially, these improvements demonstrate strong generalization capabilities, extending beyond the in-domain Dolly dataset to out-of-distribution benchmarks like Self-Instruct, SNI and Vicuna. In addition, Table 7 in Appendix G shows that student models trained with CIP exhibit higher Gini coefficients and lower entropy compared to their counterparts without CIP. This confirms that **the CIP loss can truly increase the sparsity of complex interactions**, thus effectively improving the model performance.

**Impact of the hyperparameter** $\lambda$**.** As shown in Figure 9 (a) and (b), the sparsity of complex interactions increases with $\lambda$. In contrast, the ROUGE-L score (Figure 9 (c)) and the overlap rate of complex interactions (Figure 9 (d)) initially improve with $\lambda$, but subsequently degrade as $\lambda$ continues to increase. We attribute this to the trade-off between the

two loss terms in Equation (7), where $\mathcal{L}'_{\text{CIP}}$ is designed to suppress the effects of complex interactions, while $\mathcal{L}_{\text{KD}}$ compels the student model to align the output consistency with the teacher model. When $\lambda$ is relatively small, the impact of $\mathcal{L}'_{\text{CIP}}$ is mild. At this stage, $\mathcal{L}_{\text{KD}}$ guides the student model to prioritize the suppression of task-irrelevant complex interactions while retaining salient interactions learned from the teacher model to maintain output consistency. Consequently, the overlap rate of complex interactions increases, driving an improvement in model performance. However, as $\lambda$ continues to increase, the strong impact of $\mathcal{L}'_{\text{CIP}}$ leads to a decline in the effects of salient complex interactions learned from the teacher model. Consequently, although the sparsity of complex interactions increases, the overlap rate drops significantly, leading to the degradation in model performance. Therefore, there exists an optimal range for the weight $\lambda$. By choosing a valid $\lambda$, we can strike a balance between the sparsification of interactions and the retainment of salient interactions learned from the teacher model.

*Table 2.* Comparison of ROUGE-L scores between CIP and regularization baselines.

| Model | GKD | | | | DISTILLM | | | |
|---|---|---|---|---|---|---|---|---|
| | Base | + $L_1$ | + Tf-KD | + CIP (Ours) | Base | + $L_1$ | + Tf-KD | + CIP (Ours) |
| GPT-0.3B | 26.35 | 25.18 (-1.17) | 25.88 (-0.47) | **26.52 (+0.17)** | 27.47 | 26.87 (-0.60) | 27.05 (-0.42) | **27.67 (+0.20)** |
| OPT-1.3B | 26.00 | 25.41 (-0.59) | 25.96 (-0.04) | **26.38 (+0.38)** | 27.69 | 27.13 (-0.56) | 27.33 (-0.36) | **27.77 (+0.08)** |

**Comparison with regularization baselines.** To verify that the performance gains of CIP do not merely stem from implicit regularization, we evaluate our approach against two established regularization techniques: a generic $L_1$ penalty and Tf-KD (Yuan et al., 2020), a logit-level regularization method. As shown in Table 2, applying either the $L_1$ penalty or Tf-KD consistently degrades the distillation performance. These results demonstrate that CIP provides a fundamentally effective mechanism for enhancing knowledge distillation, rather than acting merely as a regularization method.

## 6. Conclusion

In this paper, we present a comprehensive study on the mechanism of KD via interactions. We discover the common mechanism of KD is retaining salient simple interactions of the teacher model while suppressing other interactions. We further demonstrate that the performance gap among various KD methods is positively correlated with the sparsity and alignment of complex interactions. Guided by these observations, we introduce a plug-and-play loss function named CIP that explicitly suppress complex interactions. Experiments confirm that CIP generally improves various KD baselines on both in-domain and out-of-distribution benchmarks. Our study offers a new perspective on the "black box" of KD and inspires future research to utilize interactions to optimize the distillation process of LLMs.

## Acknowledgements

This work is partially supported by the Shanghai Science and Technology Commission (No. 25511102900), the National Nature Science Foundation of China (No.62376199,62576249), and the Shanghai Municipal Education Commission (No. 24CGA20).

## Impact Statement

This paper presents work whose goal is to advance the field of Machine Learning. There are many potential societal consequences of our work, none which we feel must be specifically highlighted here.

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

## A. Additional Related Work

**Knowledge distillation.** Beyond the output logits, intermediate representations contain rich information for knowledge transfer. Early works like FitNets (Romero et al., 2015) introduced "hints" to align intermediate layers. Subsequent research focused on transferring attention maps (Zagoruyko & Komodakis, 2017) or compressing specific architectures like BERT using layer-wise strategies, as seen in Patient-KD (Sun et al., 2019), MobileBERT (Sun et al., 2020), and TinyBERT (Jiao et al., 2020). More recent approaches have explored contrastive representation distillation (Tian et al., 2019) and task-aware layer-wise distillation (Liang et al., 2023b). Others have proposed homotopic distillation for pre-trained transformers (Liang et al., 2023a) or utilized Wasserstein distance to match feature distributions (Lv et al., 2024). While standard KD minimizes the KL divergence, numerous studies have sought to refine this objective. Fundamental analyses of KD's efficacy (Cho & Hariharan, 2019) and the bias-variance tradeoff (Zhou & Song, 2021) have driven improvements such as Decoupled KD (Zhao et al., 2022) and the Teacher Assistant strategy (Mirzadeh et al., 2020). To address optimization challenges, researchers have proposed multi-level logit distillation (Jin et al., 2023), logit standardization (Sun et al., 2024), and transformed teacher matching (Zheng & YANG, 2024). In the context of large language models and sequence generation, recent works have investigated $f$-divergence minimization (Wen et al., 2023), $\alpha$-$\beta$-divergence (Wang et al., 2025), and re-evaluated the role of KL divergence (Wu et al., 2025). Additionally, self-knowledge distillation via dropout (Lee et al., 2023) and pre-training distillation methods like MiniPLM (Gu et al., 2025) have been developed to enhance efficiency. Knowledge distillation has been effectively adapted for various downstream applications beyond standard model compression. In computer vision, techniques like DetKDS (Li et al., 2024) optimize distillation search specifically for object detectors. In the domain of generative language models, PromptKD (Kim et al., 2024) utilizes prompt tuning to distill student-friendly knowledge. Furthermore, distillation is applied to critical safety tasks, such as the detection and mitigation of hallucinations in large language models (Zhou et al., 2025).

## B. Masking Strategies of Input Variables

In attribution method research, it is common to employ a specific token or embedding to mask the input variables of an LLM (Lundberg & Lee, 2017; Ancona et al., 2019; Fong et al., 2019) and use changes in network outputs on the masked samples to estimate attributions of different input variables. The selection of a masking approach is complex, as each method has its weakness. For example, replacing input variables with the mean baseline value (the average of all samples) or the zero baseline value can introduce out-of-distribution signals, thereby providing the model with artificial information, such as uniform grey or black dots in an image (Dabkowski & Gal, 2017; Ancona et al., 2019; Sundararajan et al., 2017). Additionally, blurring image pixels using a Gaussian kernel (Fong & Vedaldi, 2017; Fong et al., 2019) as the masked state removes high-frequency signals but fails to eliminate low-frequency signals (Covert et al., 2021; Sturmfels et al., 2020).

Given these challenges, we adopt a token replacement strategy, which is standard for the text domain. This involves substituting the target input word with a dedicated [MASK] token at the embedding level. For example, to mask the word "green" in the input "He is a green hand," we would provide the LLM with the modified input "He is a [MASK] hand." This approach effectively nullifies the specific semantic contribution of the target word without introducing out-of-distribution artifacts, ensuring a clean and consistent baseline for our interaction analysis. For the specific [MASK] token for each LLM, please refer to Section 5 for details.

## C. Details about How to Compute $v(x)$ for Text Generation Tasks

Prior studies typically calculate $v(x)$ based on the probability of generating a single next token. For example, given an LLM $v$ and an input sentence $\boldsymbol{x}$ with $n$ input words indexed by $N = \{1, 2, \ldots, n\}$, let $v(\boldsymbol{x}) \in \mathbb{R}$ denote the scalar output of the LLM. Here we set $v(\boldsymbol{x}) = \log \frac{p(y=y^*|\boldsymbol{x})}{1-p(y=y^*|\boldsymbol{x})} \in \mathbb{R}$, where $p(y = y^*|\boldsymbol{x})$ represents the probability of generating the ground truth token $y^*$ given the input $\boldsymbol{x}$.

However, this token-level approach fails to capture the complex reasoning process inherent in generating long sentences, making it insufficient for standard text generation tasks. To address this limitation, we extend the definition of $v(x)$ from a single-token basis to a sequence-level metric.

For text generation task and given the ground truth sequence $y^* = (y_1^*, \ldots, y_L^*)$, we define $\bar{p}$ as the geometric mean of the probabilities of each ground truth token in $y^*$, *i.e.*, $\bar{p} = \left( \prod_{l=1}^{L} p(y_l^*|x, y_{<l}^*) \right)^{1/L}$ Then scalar output $v(x)$ is defined as $v(x) = \log \frac{\bar{p}}{1-\bar{p}} \in \mathbb{R}$. To ensure numerical stability, we clip $\bar{p}$ to $[\epsilon, 1-\epsilon], \epsilon = 10^{-7}$ in our implementation.

Our choice of the scalar output function $v(x)$ is theoretically grounded in the standard evaluation metric for generative models: Perplexity (PPL). Formally, the perplexity of a sequence is defined as $\text{PPL} = \left(\prod_{l=1}^{L} p(y_l^* | x, y_{<l}^*)\right)^{-1/L}$. Our geometric mean probability $\bar{p}$ is strictly the inverse of perplexity, *i.e.*, $\bar{p} = \text{PPL}^{-1}$, representing the length-normalized average likelihood assigned to the ground truth. To satisfy the Universal Matching Property of Harsanyi interactions which requires the output space to be unbounded (in $\mathbb{R}$ rather than $[0, 1]$), we apply the log-odds transformation to $\bar{p}$. Thus, $v(x)$ quantifies the model's confidence in generating the precise ground truth sequence, normalized by length and mapped to a theoretically rigorous additive space.

# D. Proof of Theorem

### D.1. Proof of Universal Matching Property

In this section, we provide the formal proof for the Universal Matching Property of the complete AND interaction framework. We will demonstrate that the output of the surrogate logical model, which is the sum of all AND interaction effects, can perfectly match the output of the LLM for any masked sample.

The **surrogate logical model** $\phi(\cdot)$ is defined as follows:

$$\phi(x_T) \triangleq \phi(x_\emptyset) + \sum_{S \subseteq N, S \neq \emptyset} \mathbb{1}_{\text{AND}}(S \mid x_T) \cdot I_S^{\text{AND}} \tag{8}$$

where the AND trigger function $\mathbb{1}_{\text{AND}}(S \mid x_T) \in \{0, 1\}$ represents an **AND relationship** between input variables in $S$, which can also be termed **AND interaction pattern**. The scalar weight $I_S^{\text{AND}}$ quantifies the effect of an AND relationship, which can also be termed **AND interaction effect**. An AND relationship is activated only by the joint presence of all input variables in the set $S$, *i.e.*, all input variables in $S$ are not masked. For instance, given the input sentence $x = $ *"He is a green hand,"* the co-occurrence of the input variables in the set $S = \{green, hand\}$ contributes a numerical effect $I_S^{\text{AND}}$ that pushes the surrogate logical model's inference towards the semantic meaning of *"beginner."* If an AND interaction $S$ is triggered, *i.e.*, $\mathbb{1}_{\text{AND}}(S \mid x_T) = 1$, the corresponding interaction effect $I_S^{\text{AND}}$ is added to the output of the logical model. Otherwise, if any word in $S$ is masked and the AND interaction is not triggered, *i.e.*, $\mathbb{1}_{\text{AND}}(S \mid x_T) = 0$, the interaction effect $I_S^{\text{AND}}$ is not added to the output of the logical model. $x_\emptyset$ represents that all input variables in $N$ are masked.

**Definition of universal matching property for AND interactions.** When the scalar weights in the surrogate logical model $\phi(\cdot)$ are set to $I_S^{\text{AND}} = \sum_{S' \subseteq S}(-1)^{|S|-|S'|} v_{\text{and}}(x_{S'})$, the output of $\phi(\cdot)$ can always match the output score of the LLM $v(\cdot)$, *i.e.*, $\forall T \subseteq N, v(x_T) = \phi(x_T)$. Here $v_{\text{and}}(x_T) = v(x_T)$.

We need to prove that given an input sample $x$, for each masked sample $\{x_T | T \subseteq N\}$, the network output score $v(x_T) \in \mathbb{R}$ can be well matched by the surrogate logical model $\phi(x_T)$. The surrogate logical model $\phi(x_T)$ uses the sum of AND interactions to accurately explain/match the network output score $v(x_T)$.

$$\forall T \subseteq N, v(x_T) = \phi(x_T).$$
$$\phi(x_T) = \phi(x_\emptyset) + \sum_{S \subseteq N, S \neq \emptyset} \mathbb{1}_{\text{AND}}(S \mid x_T) \cdot I_S^{\text{AND}}$$
$$= \underbrace{v(x_\emptyset) + \sum_{S \subseteq T, S \neq \emptyset} I_S^{\text{AND}}}_{v_{\text{and}}(x_T)} \tag{9}$$

*Proof.* **Universal matching property of AND interactions.** For all $2^n$ masked samples $\{x_T \mid T \subseteq N\}$, what we need to prove is that the output $v_{\text{and}}(x_T)$ of an LLM can be universally explained by all the interactions in $T \subseteq N$, *i.e.*, $\forall S \subseteq T, S \neq \emptyset, v_{\text{and}}(x_T) = \sum_{S \subseteq T, S \neq \emptyset} I_S^{\text{AND}}(x) = v(x_\emptyset) + \sum_{S \subseteq T, S \neq \emptyset} I_S^{\text{AND}}$. Here, $v(x_\emptyset) = v_{\text{and}}(x_\emptyset)$.

According to the definition of the AND interaction, $I_S^{\text{AND}}(x) = \sum_{L \subseteq S}(-1)^{|S|-|L|} v_{\text{and}}(x_L)$. To simplify the computation of the sum of AND interactions $\sum_{S \subseteq T, S \neq \emptyset} I_S^{\text{AND}}(x) = \sum_{S \subseteq T, S \neq \emptyset} \sum_{L \subseteq S}(-1)^{|S|-|L|} v_{\text{and}}(x_L)$, we exchange the order of summation of the set $L \subseteq S \subseteq T$ and the set $S \supseteq L$. Given a set of input variables $L$, we compute all linear combinations of all sets $S$ containing $L$ with respect to the model outputs $v_{\text{and}}(x_S)$, *i.e.*, $\sum_{S:L \subseteq S \subseteq T}(-1)^{|S|-|L|} v_{\text{and}}(x_L)$. Then, we compute all summations over the set $L \subseteq T$ as $\sum_{S \subseteq T, S \neq \emptyset} I_S^{\text{AND}}(x) = \sum_{L \subseteq T} \sum_{S:L \subseteq S \subseteq T}(-1)^{|S|-|L|} v_{\text{and}}(x_L) - v_{\text{and}}(x_\emptyset)$. Then, we can compute different cases of $L \subseteq S \subseteq T$ as follows:

(1) When $L = T = S$, $\sum_{S:L \subseteq S \subseteq T} (-1)^{|S|-|L|} v_{\text{and}}(\boldsymbol{x}_L) = (-1)^{|T|-|T|} v_{\text{and}}(\boldsymbol{x}_L) = v_{\text{and}}(\boldsymbol{x}_L)$.

(2) When $L \subseteq S \subseteq T, L \neq T$, let us consider the linear combinations of all sets $S$ with number $|S|$ for the model output $v_{\text{and}}(\boldsymbol{x}_L)$, respectively. Let $m := |S| - |L|$, $(0 \leq m \leq |T| - |L|)$, then there are a total of $C_{|T|-|L|}^m$ combinations of all sets $S$ of order $|S|$. Given $L$, accumulating the model outputs $v_{\text{and}}(\boldsymbol{x}_L)$ corresponding to all $S \supseteq L$, we can get

$$\sum_{S:L \subseteq S \subseteq T} (-1)^{|S|-|L|} v_{\text{and}}(\boldsymbol{x}_L) = v_{\text{and}}(\boldsymbol{x}_L) \cdot \underbrace{\sum_{m=0}^{|T|-|L|} C_{|T|-|L|}^m (-1)^m}_{=0} = 0.$$

Considering all the cases, the complete derivation of the sum of AND interactions is as follows.

$$
\begin{aligned}
&\sum_{S \subseteq T, S \neq \emptyset} I_S^{\text{AND}} \\
&= \sum_{S \subseteq T, S \neq \emptyset} \sum_{L \subseteq S} (-1)^{|S|-|L|} v_{\text{and}}(\boldsymbol{x}_L) \\
&= \sum_{L \subseteq T} \sum_{S:L \subseteq S \subseteq T} (-1)^{|S|-|L|} v_{\text{and}}(\boldsymbol{x}_L) - v_{\text{and}}(\boldsymbol{x}_\emptyset) \\
&= \underbrace{v_{\text{and}}(\boldsymbol{x}_T)}_{L=T} + \sum_{L \subseteq T, L \neq T} v_{\text{and}}(\boldsymbol{x}_L) \cdot \underbrace{\sum_{m=0}^{|T|-|L|} C_{|T|-|L|}^m (-1)^m}_{=0} - v_{\text{and}}(\boldsymbol{x}_\emptyset) \\
&= v_{\text{and}}(\boldsymbol{x}_T) - v(\boldsymbol{x}_\emptyset)
\end{aligned}
\tag{10}
$$

Therefore, we have proved that $\forall \emptyset \neq T \subseteq N, v_{\text{and}}(\boldsymbol{x}_T) = v(\boldsymbol{x}_\emptyset) + \sum_{S \subseteq T, S \neq \emptyset} I_S^{\text{AND}}$. We can easily get $v(x_T) = \phi(x_T) = v_{\text{and}}(x_T) = v(x_\emptyset) + \sum_{S \subseteq T, S \neq \emptyset} I_S^{\text{AND}}$, thus, we obtain the universal matching property of AND interactions.

$\square$

### D.2. Proof of Partition Size Constraint for the Sampling Method

**Proposition.** *To ensure that any non-empty union of the partitioned subsets represents a complex interaction, the number of partitions $m$ must satisfy $m \leq 3$.*

*Proof.* Given an input sequence $x$ with $n$ words indexed by $N = \{1, \ldots, n\}$, and let $\mathcal{P} = \{S_1, S_2, \ldots, S_m\}$ be a partition of $N$ such that $\bigcup_{i=1}^m S_i = N$ and $S_i \cap S_j = \emptyset$ for $i \neq j$. We define the set of complex interactions as $\Omega^{\text{complex}} = \{S \subseteq N \mid \lceil n/3 \rceil \leq |S| \leq n\}$.

To guarantee that the sampling strategy is valid, we require that any non-empty combination of these subsets constitutes a complex interaction. Formally, this condition is expressed as:

$$\forall K \subseteq \{1, \ldots, m\}, K \neq \emptyset \implies \bigcup_{i \in K} S_i \in \Omega^{\text{complex}} \tag{11}$$

Substituting the definition of $\Omega^{\text{complex}}$, this requires:

$$\left| \bigcup_{i \in K} S_i \right| \geq \lceil n/3 \rceil \tag{12}$$

Since this condition must hold for *all* valid $K$, it must specifically hold for the singleton sets where $|K| = 1$. Let $K = \{i\}$ for any $i \in \{1, \ldots, m\}$. The condition implies a lower bound on the size of each individual subset:

$$|S_i| \geq \lceil n/3 \rceil, \quad \forall i \in \{1, \ldots, m\} \tag{13}$$

On the other hand, since $\{S_i\}$ forms a disjoint partition of $N$, the sum of their cardinalities must equal the total number of words $n$:

$$\sum_{i=1}^m |S_i| = |N| = n \tag{14}$$

Combining this with the inequality in Eq. (13), we obtain:

$$\sum_{i=1}^{m} \lceil n/3 \rceil \leq \sum_{i=1}^{m} |S_i| = n \tag{15}$$

Simplifying the summation:

$$m \cdot \lceil n/3 \rceil \leq n \tag{16}$$

Solving for $m$:

$$m \leq \frac{n}{\lceil n/3 \rceil} \tag{17}$$

By the definition of the ceiling function, we know that $\lceil n/3 \rceil \geq n/3$. Therefore:

$$\frac{n}{\lceil n/3 \rceil} \leq \frac{n}{n/3} = 3 \tag{18}$$

Consequently, we derive the strict upper bound $m \leq 3$. This proves that to strictly satisfy the definition of complex interactions for all sampled combinations, the partition size $m$ is mathematically constrained to be a small constant (specifically 2 or 3). □

## E. Implementation Details of Analysis

To construct the set of input variables $N$ for our interaction analysis, we extracted meaningful words from the "Instruction" segment of each input. Following the preprocessing protocol of Shen et al. (2023), we defined meaningful words as tokens that are neither punctuation marks nor stop words listed in the NLTK library (Bird, 2006). We standardized the number of input variables at $n = |N| = 13$. As detailed in Appendix F.3, the instructions in the `databricks-dolly-15k` dataset are usually short; thus, a set size of $n = 13$ is empirically sufficient to encompass the core semantic content of the samples. During the masking process, we exclusively applied masking to the selected tokens within $N$, while leaving all other "background" tokens unchanged.

### E.1. Baselines

- **KD** (Hinton et al., 2015) minimizes the forward Kullback-Leibler Divergence (FKLD) between the teacher and student distributions using the fixed training dataset.

- **SeqKD** (Kim & Rush, 2016) maximizes the likelihood of high-probability sequences generated by the teacher, effectively functioning as Supervised Fine-Tuning (SFT) on teacher-generated outputs.

- **ImitKD** (Lin et al., 2020) addresses the training-inference mismatch by introducing Student-Generated Outputs (SGO). It optimizes the standard KLD loss on sequences generated by the student model, forcing the student to mimic the teacher's behavior on its own exploration path.

- **MiniLLM** (Gu et al., 2024) also utilizes SGOs but employs an on-policy gradient method. It aims to minimize the Reverse KLD (RKLD) to prevent the student from assigning high probabilities to low-probability teacher regions (mode collapse).

- **GKD** (Agarwal et al., 2024) adopts the Generalized Jensen-Shannon Divergence (JSD) as the objective. It trains on a mixture of datasets, combining fixed data (ground truth or teacher-generated) with on-policy student-generated sequences.

- **DistiLLM** (Ko et al., 2024) introduces the Skew KL (or Skew RKL) divergence to stabilize training. It utilizes a mixed dataset and proposes an adaptive off-policy method to selectively incorporate SGOs based on validation loss, thereby filtering out noisy generated data.

# F. Implementation Details of Training

## F.1. Training Details

For training the teacher and student models, we used eight RTX PRO 6000 96GB GPUs. Our experimental setup for training LMs on `databricks-dolly-15k` primarily follows the experimental setup for Ko et al. (2024). For models within 1B parameters, we search for the learning rates in {5e-4, 1e-4, 5e-5}, the batch sizes in {8, 16, 32} and train these models for 20 epochs. For models that have more than 1B parameters, we search for the learning rate in {5e-5, 1e-5, 5e-6}, the batch sizes of {8, 16, 32}, and train these models for 10 epochs. We fully use the distillation loss for the instruction-following dataset and language modeling loss for OpenWebText (Gokaslan et al., 2019) corpus. The checkpoints of each student are selected by the ROUGE-L scores on the validation set. To ensure the effectiveness of CIP, specifically when employing Student-Generated Outputs (SGO), we calculate CIP based on the ground truth from the fixed dataset. Furthermore, we only calculated CIP for samples where the number of input variables in the "Instruction" segment, maintaining consistency with the settings used in our interaction analysis.

## F.2. Evaluation

For evaluating the teacher and student models, we applied two RTX PRO 6000 96GB GPUs. Following Gu et al. (2024); Ko et al. (2024), we adopt a prompt template as shown in Figure 10. We sample the responses from each model using a temperature of 1.0, a max-length limit of 512. For GPT-5 feedback, we use a popular prompt introduced in Zheng et al. (2023) which is illustrated in Figure 11, the prompt asking GPT-5 to compare model-generated responses with the ground truth answers and give 1-10 scores for both responses. We report the ratio of the total score of model responses and ground truth answers by following Ko et al. (2024).

> Below is an instruction that describes a task.
> Write a response that appropriately completes the request.
>
> ### Instruction:
> {instruction}
>
> ### Input:
> {input}
>
> ### Response:

*Figure 10.* The prompt template for training and evaluation of instruction-following task experiments.

> [System]
> Please act as an impartial judge and evaluate the quality of the response provided by an AI assistant to the user question displayed below. Your evaluation should consider factors such as the helpfulness, relevance, accuracy, depth, creativity, and level of detail of the response. Begin your evaluation by providing a short explanation. Be as objective as possible. After providing your explanation, please rate the response on a scale of 1 to 10 by strictly following this format: "[[rating]]", for example: "Rating: [[5]]".
>
> [Question]
> {question}
>
> [The Start of Assistant's Answer]
> {answer}
> [The End of Assistant's Answer]

*Figure 11.* The prompt template for single-answer grading of GPT-5 feedback from Zheng et al. (2023).

## F.3. Dataset Description

- **databricks-dolly-15k** (Conover et al., 2023): This is an open-source dataset comprising instruction-following records generated by Databricks employees. It encompasses several behavioral categories outlined in (Ouyang et al., 2022), including brainstorming, classification, closed/open QA, generation, information extraction, and summarization. Some specific examples provided in Table 3.

- **Self-Instruct** (Wang et al., 2023): SelfInstruct is a framework designed to enhance the language model's instruction-following capabilities by leveraging the model's own outputs to generate a vast set of instructional data. It consists of 52k instructions and 82k instance inputs/outputs for fine-tuning, supplemented by 252 expert-written tasks for practical applications and an additional 50k examples from public datasets for benchmarking.

- **Vicuna** (Chiang et al., 2023): Following the evaluation protocol of (Ko et al., 2024), we incorporate the set of 80 challenging questions originally curated for evaluating the Vicuna model to assess model performance on complex queries.

- **Super-Natural Instructions** (Wang et al., 2022): This benchmark features a diverse collection of 1,616 NLP tasks paired with expert-written instructions, spanning 76 distinct task types. For evaluation, we utilize its test set, which comprises approximately 9k samples drawn from 119 different tasks.

*Table 3.* Some cases from the `databricks-dolly-15k` test set.

| Field | Content |
|---|---|
| **Case #1** | |
| **Topic** | open QA |
| **Instruction** | What is the name of the largest red-light district in Amsterdam? |
| **Input Variables** | ["What", "is", "the", "name", "of", "the", "largest", "red-light", "district", "in", "Amsterdam"] |
| **Input** | None |
| **Ground Truth** | The largest red-light district in Amsterdam is De Wallen. |
| **Case #2** | |
| **Topic** | closed QA |
| **Instruction** | What is the old name for The Go Between Bridge? |
| **Input Variables** | ["What", "is", "the", "old", "name", "for", "The", "Go", "Between", "Bridge"] |
| **Input** | The Go Between Bridge, formerly known as the Hale Street Link, is a toll bridge for vehicles, pedestrians and cyclists over the Brisbane River in inner-city Brisbane, Queensland, Australia. The bridge connects Merivale and Cordelia Streets in West End to Hale Street and the Inner City Bypass at Milton. It is Brisbane's first inner-city traffic bridge in 40 years. |
| **Ground Truth** | Hale Street Link |
| **Case #3** | |
| **Topic** | classification |
| **Instruction** | Identify which car manufacturer is German or American: Opel, GMC? |
| **Input Variables** | ["Identify", "which", "car", "manufacturer", "is", "German", "or", "American:", "Opel", "GMC"] |
| **Input** | None |
| **Ground Truth** | Opel is German, GMC is American |

# G. More Experiment Results

## G.1. Comparison of the training time with and without CIP loss

*Table 4.* Comparison of the training time per batch (in seconds/batch) between different KD methods without and with the CIP loss function across various model sizes. The results are averaged over 20 epochs. The overhead is calculated as the increase in time relative to the original method.

| Model | Method | w/o CIP (s/batch) | w/ CIP (s/batch) | Increased Time (s/batch) | Increased Time (%) |
|---|---|---|---|---|---|
| GPT-2-0.1B | KD | 0.318 | 0.354 | 0.036 | 11.32 |
| | SeqKD | 0.331 | 0.359 | 0.028 | 8.46 |
| | ImitKD | 2.049 | 2.126 | 0.077 | 3.76 |
| | MiniLLM | 2.659 | 2.741 | 0.082 | 3.08 |
| | GKD | 1.675 | 1.745 | 0.070 | 4.18 |
| | DistiLLM | 0.632 | 0.670 | 0.038 | 6.01 |
| GPT-2-0.3B | KD | 0.654 | 0.710 | 0.056 | 8.56 |
| | SeqKD | 0.668 | 0.732 | 0.064 | 9.58 |
| | ImitKD | 3.801 | 3.940 | 0.139 | 3.66 |
| | MiniLLM | 4.913 | 5.062 | 0.149 | 3.03 |
| | GKD | 3.148 | 3.277 | 0.129 | 4.10 |
| | DistiLLM | 1.262 | 1.333 | 0.071 | 5.63 |

To assess the computational cost introduced by our proposed CIP loss, we compared the average training time per batch of various distillation methods with and without the CIP loss. The experiments were conducted on a server equipped with 4 × RTX PRO 6000 96GB GPUs. We measured the average training time (s/batch) over 20 epochs for both GPT-2-0.1B and GPT-2-0.3B student models. As shown in Table 4, the integration of CIP introduces a marginal computational overhead, ranging from approximately 3.03% to 11.32% across different methods and model sizes. This slight increase is primarily attributed to the sampling of subsets and the calculation of interaction terms. Considering the significant performance improvements demonstrated in the main experiments, we consider this computational cost to be negligible and acceptable for practical applications.

## G.2. More Results of the Main Experiments

Figure 12 illustrates the distributions of Gini coefficient $G(x)$ and Shannon entropy $H(x)$ across all samples in the test set. Results show that distilled student models demonstrate an obvious distributional shift towards higher Gini values and lower entropy compared to corresponding base models and SFT models.

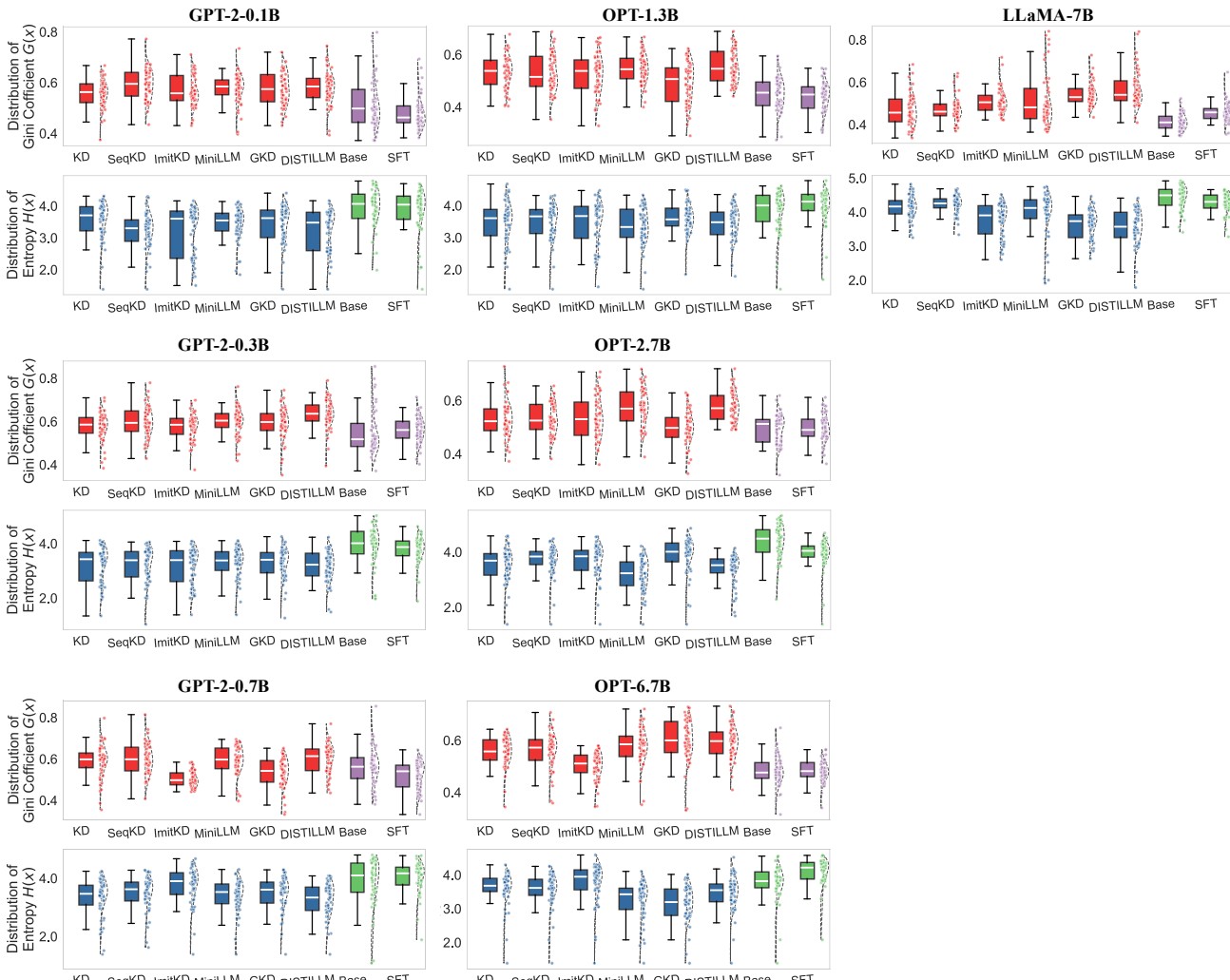

*Figure 12.* The distribution of the Gini coefficient and entropy across varying samples.

Table 5 shows that distilled student models exhibit a higher overlap rate with the teacher model in terms of salient interactions than that of the non-distilled student models.

*Table 5.* Comparing the student-teacher overlap rate of salient interactions before and after distillation. $\overline{Overlap@k} = \mathbb{E}_x\left[Overlap@k(x)\right]$ represents the average overlap rate across all samples.

| Metric | Model | Non-distilled | Distilled | | | | | |
|---|---|---|---|---|---|---|---|---|
| | | Base | KD | SeqKD | MiniLLM | ImitKD | GKD | DISTILLM |
| $k = 0.05$ | GPT-2-0.1B | 11.79% | 18.62% | 19.57% | 17.37% | 14.97% | 17.05% | 18.89% |
| | GPT-2-0.3B | 14.23% | 20.84% | 19.99% | 22.00% | 19.06% | 19.23% | 23.14% |
| | GPT-2-0.7B | 18.96% | 21.89% | 22.14% | 25.22% | 18.15% | 23.10% | 23.20% |
| | OPT-1.3B | 17.30% | 21.58% | 23.26% | 21.58% | 16.26% | 22.04% | 17.51% |
| | OPT-2.7B | 19.33% | 22.48% | 22.71% | 23.70% | 19.75% | 20.59% | 25.45% |
| | OPT-6.7B | 20.78% | 21.91% | 22.87% | 25.21% | 21.04% | 24.05% | 24.94% |
| | LLaMA-7B | 15.04% | 18.09% | 16.02% | 20.22% | 14.65% | 20.72% | 23.36% |
| $k = 0.1$ | GPT-2-0.1B | 19.81% | 24.84% | 26.96% | 23.68% | 23.05% | 26.69% | 26.38% |
| | GPT-2-0.3B | 22.99% | 30.54% | 27.27% | 31.61% | 25.65% | 27.45% | 29.12% |
| | GPT-2-0.7B | 24.73% | 28.40% | 29.28% | 32.86% | 26.31% | 31.00% | 32.98% |
| | OPT-1.3B | 21.96% | 25.24% | 26.48% | 26.54% | 21.47% | 25.23% | 25.71% |
| | OPT-2.7B | 23.06% | 26.90% | 27.75% | 31.04% | 26.68% | 26.20% | 31.40% |
| | OPT-6.7B | 24.93% | 26.37% | 29.43% | 33.17% | 25.23% | 30.69% | 32.22% |
| | LLaMA-7B | 21.35% | 24.18% | 21.55% | 27.62% | 22.02% | 29.43% | 31.60% |
| $k = 0.15$ | GPT-2-0.1B | 25.45% | 30.14% | 32.91% | 30.76% | 29.58% | 33.09% | 33.05% |
| | GPT-2-0.3B | 27.84% | 35.68% | 33.79% | 36.97% | 33.47% | 34.95% | 34.80% |
| | GPT-2-0.7B | 30.21% | 33.44% | 33.43% | 38.06% | 31.61% | 34.37% | 38.27% |
| | OPT-1.3B | 25.75% | 30.67% | 33.02% | 31.33% | 26.82% | 28.98% | 31.44% |
| | OPT-2.7B | 27.50% | 31.94% | 32.49% | 34.80% | 31.08% | 30.58% | 36.22% |
| | OPT-6.7B | 30.59% | 32.14% | 32.88% | 37.52% | 31.54% | 35.45% | 35.32% |
| | LLaMA-7B | 27.57% | 29.02% | 26.27% | 31.83% | 28.19% | 33.95% | 35.63% |

Figure 13 plots the change of Gini coefficients between distilled and base models for simple and complex interactions. Results show that the sparsity of complex interactions exhibits an obvious increase after distillation.

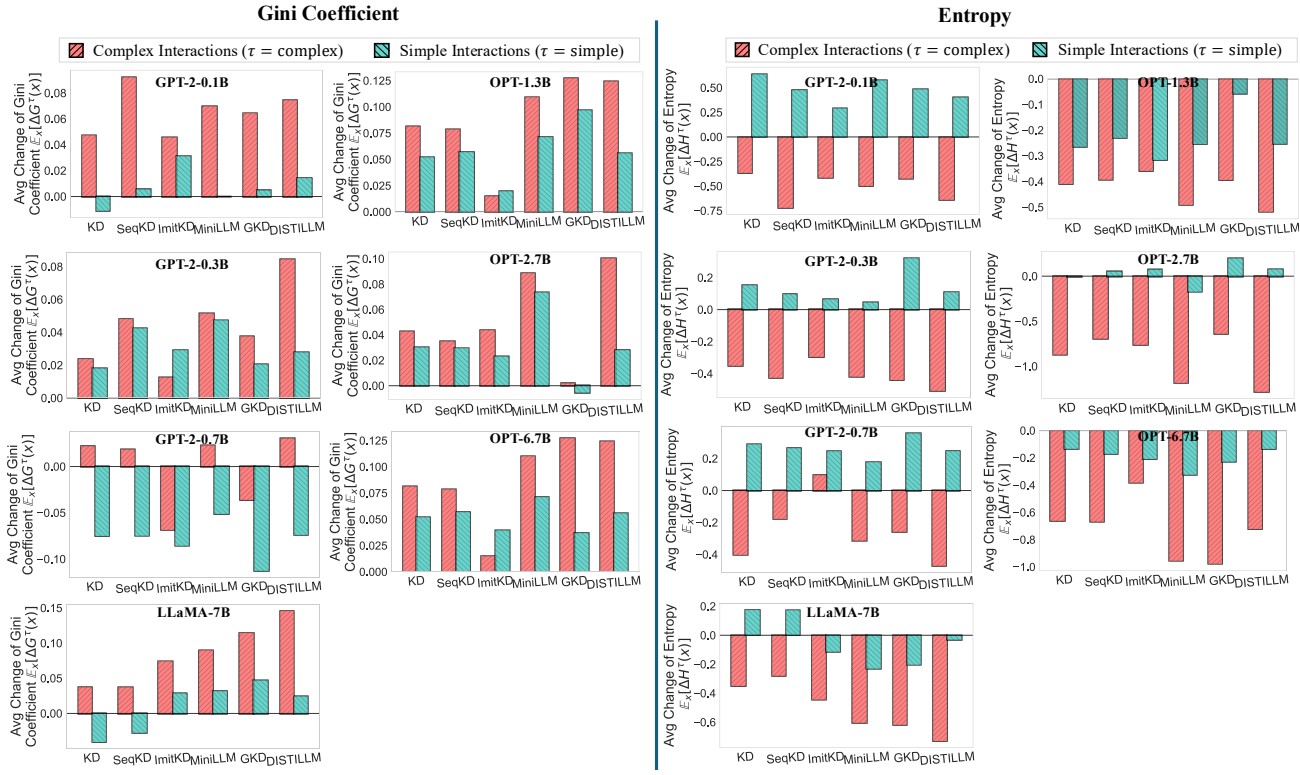

*Figure 13.* Comparison of sparsity changes in simple versus complex interactions between non-distilled and distilled models.

Figure 14 shows the overlap rate of salient simple interactions consistently surpasses that of complex interactions. This indicates that the student model learn more simple salient interactions than complex salient interactions from the teacher model.

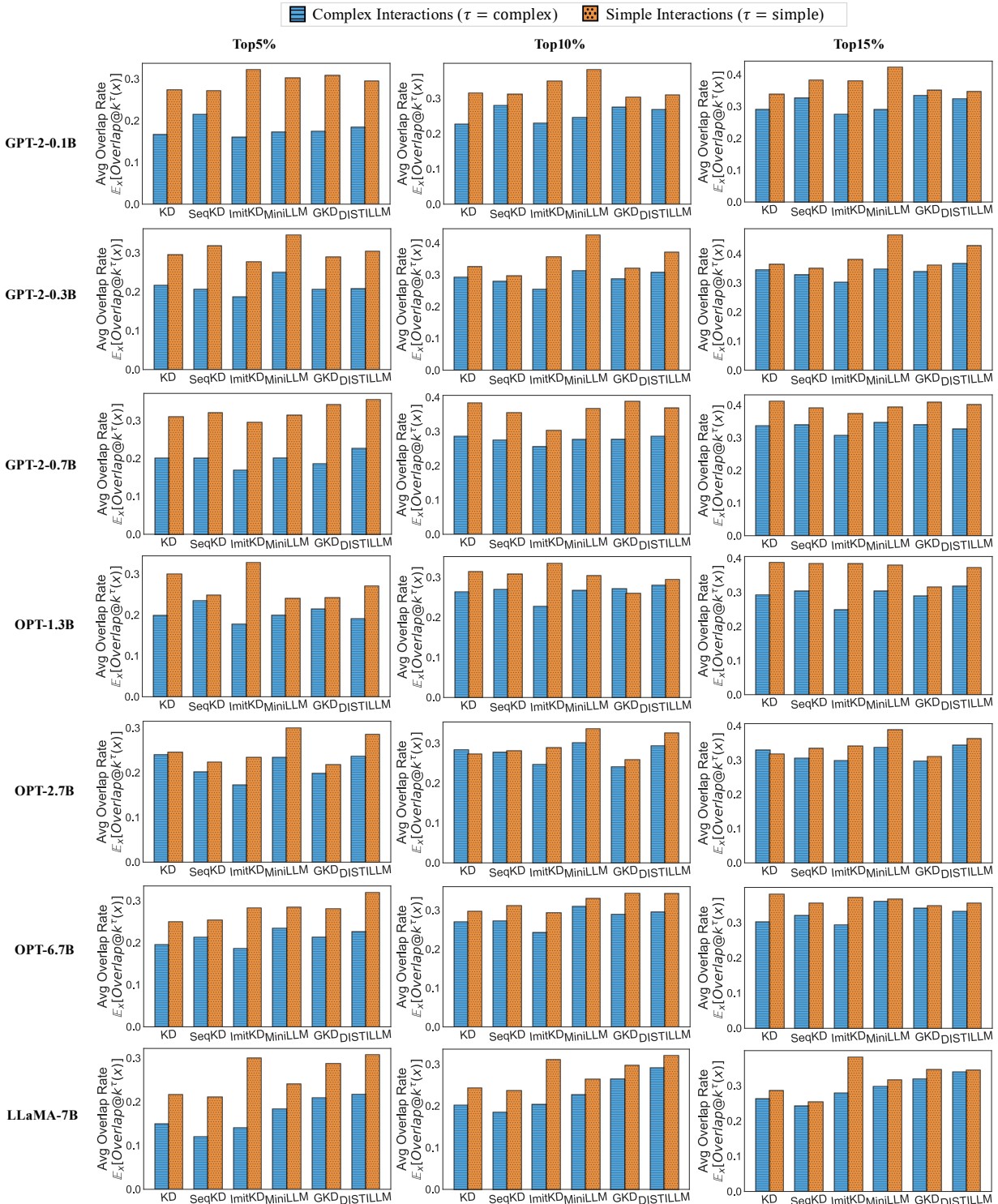

*Figure 14.* Comparison of overlap rate between simple interactions and complex interactions after distillation.

Figure 15 shows that the sparsity of complex interactions is positively correlated with model performance. Student models that exhibit higher Gini coefficients and lower entropy (indicating higher sparsity) in their complex interactions generally achieve superior ROUGE-L scores.

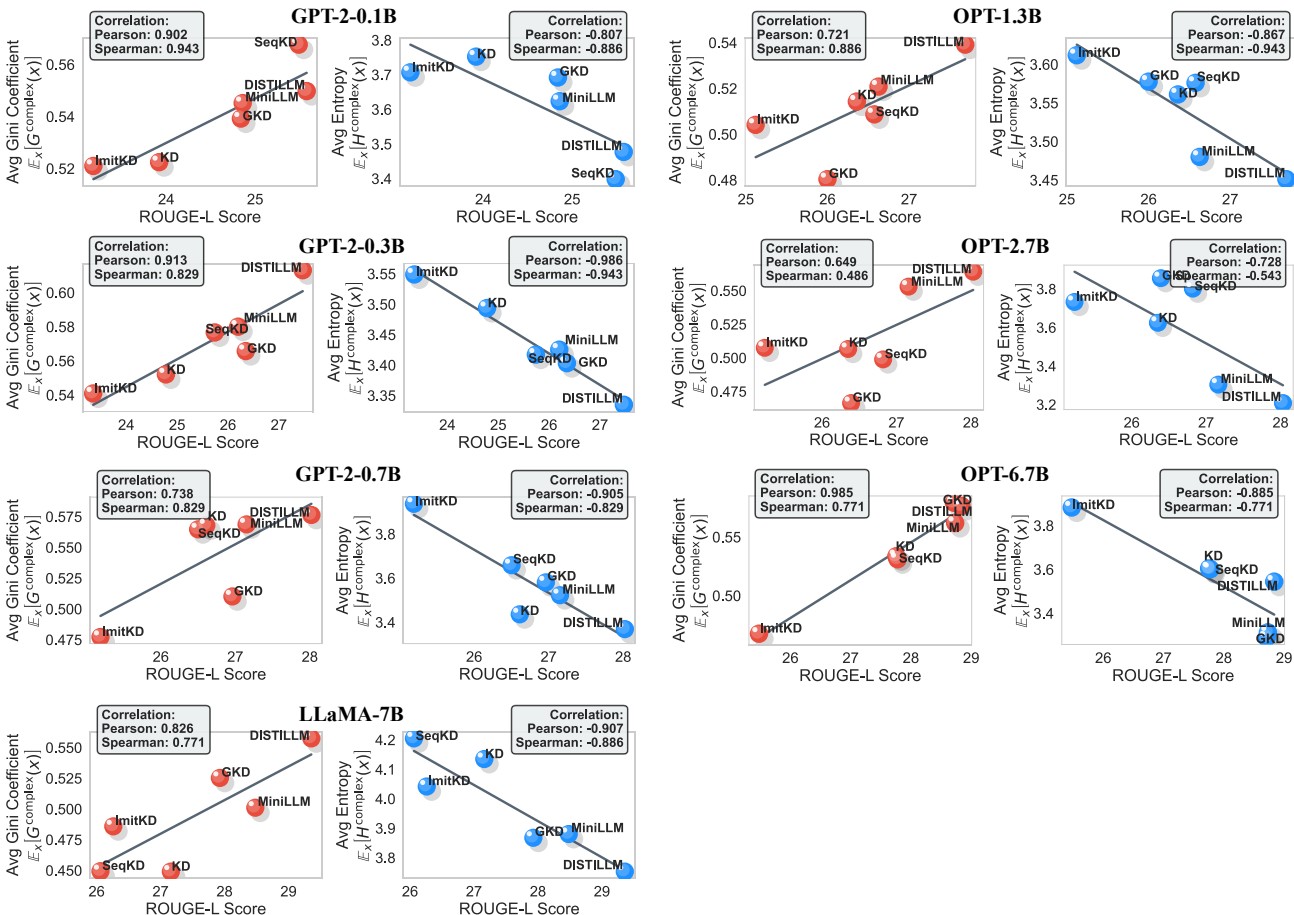

*Figure 15.* Correlation between the sparsity of complex interactions and model performance.

Figure 16 shows that there is no distinct relationship between the sparsity of simple interactions and the model performance.

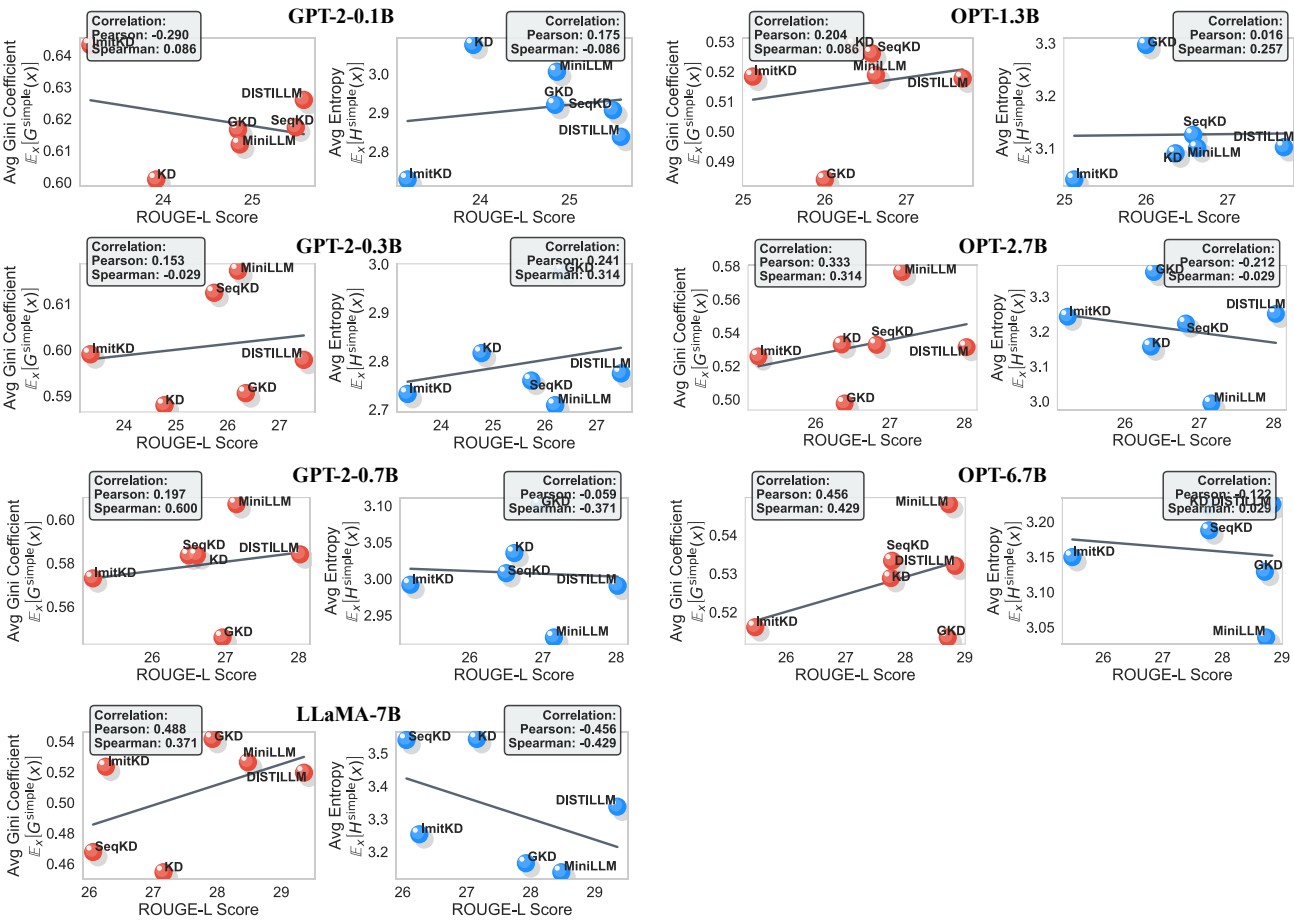

*Figure 16.* Correlation between the sparsity of simple interactions and model performance.

As shown in Figure 17, the results reveal a positive correlation between model performance and the overlap rate of complex interactions. This indicates that student models with higher performance learn more complex interactions from the teacher model compared to student models with lower performance. Counter-intuitively, the overlap rate of simple interactions shows no distinct correlation with model performance, as shown in Figure 18.

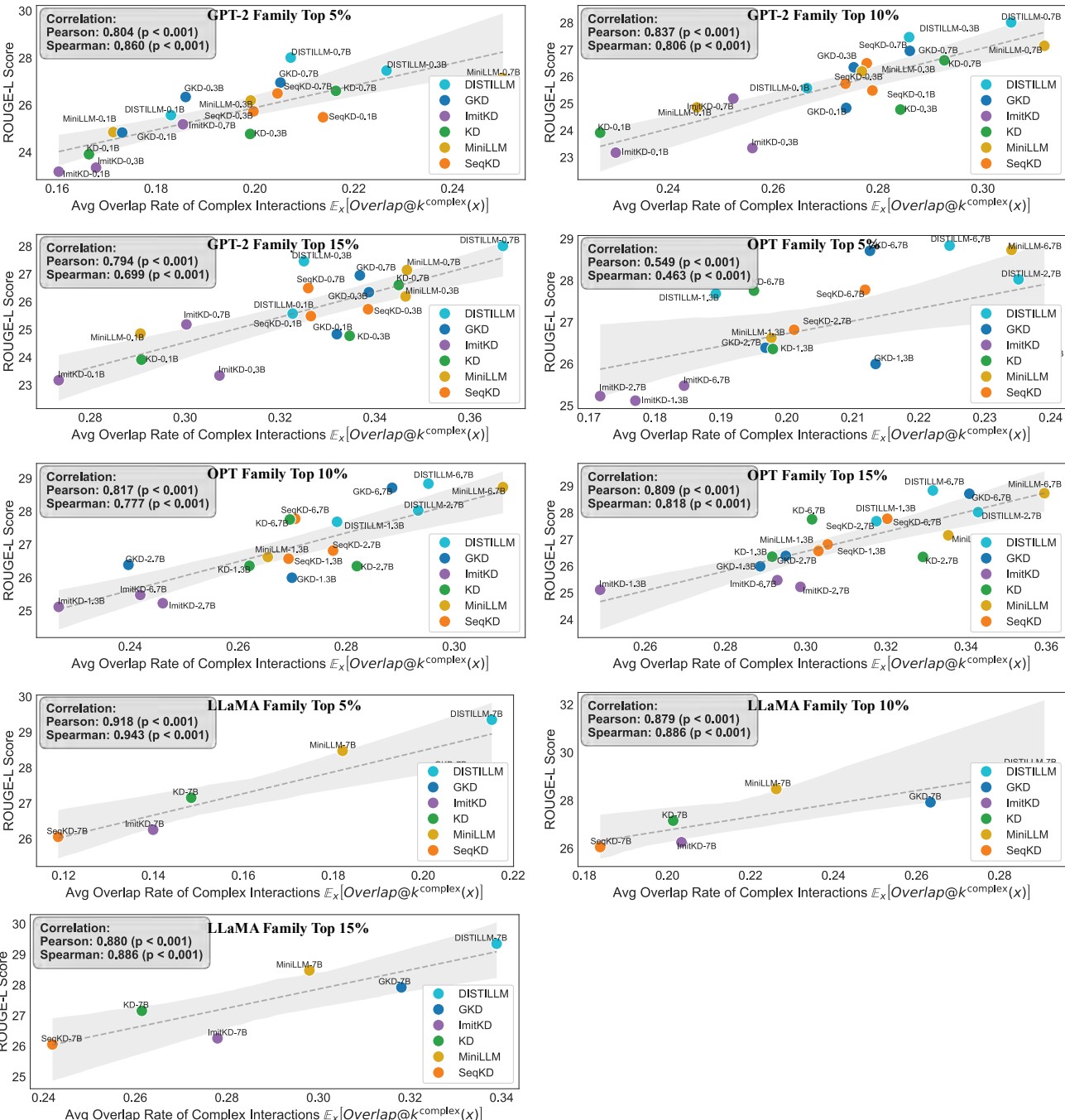

*Figure 17.* Correlation between the student-teacher overlap rate of complex interactions and model performance.

As shown in Figure 18, simple interactions shows no distinct correlation with model performance.

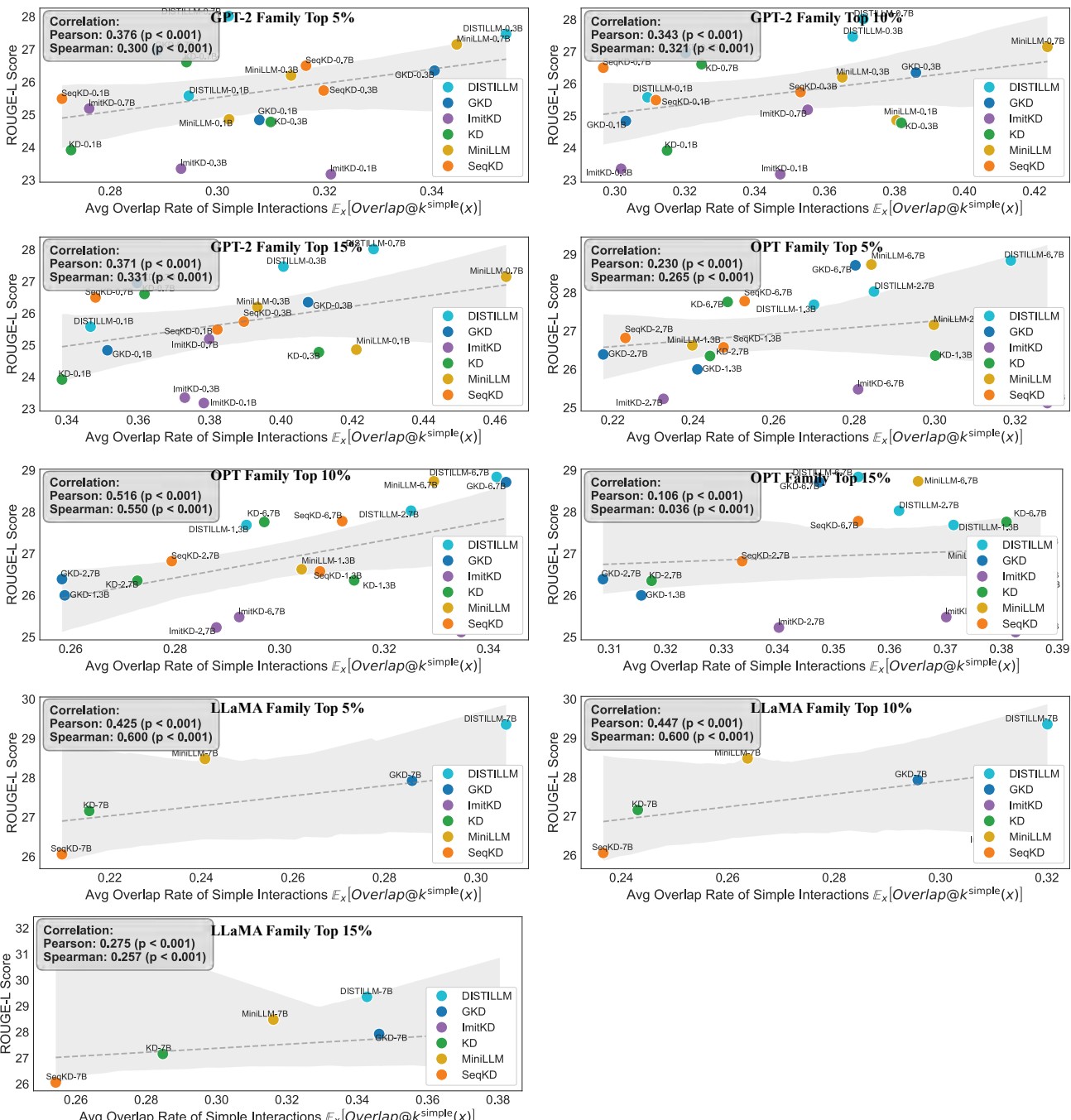

*Figure 18.* Correlation between the student-teacher overlap rate of simple interactions and model performance.

*Table 6.* The comparison of ROUGE-L and GPT-5 Eval score between different KD methods without and with the CIP loss function across various model families and model sizes. Results show that adding CIP consistently improves the performance of all original KD methods.

| Model | #Params | Method | Dolly | | | | SelfInst | | | | Vicuna | | | | SNI | |
| --- | --- | --- | --- | --- | --- | --- | --- | --- | --- | --- | --- | --- | --- | --- | --- | --- |
| | | | Rouge-L | | GPT-5 Eval | | Rouge-L | | GPT-5 Eval | | Rouge-L | | GPT-5 Eval | | Rouge-L | |
| | | | w/o | w/ CIP | w/o | w/ CIP | w/o | w/ CIP | w/o | w/ CIP | w/o | w/ CIP | w/o | w/ CIP | w/o | w/ CIP |
| GPT-2 | 1.5B | Teacher | 26.42 | - | 47.33 | - | 14.65 | - | 32.01 | - | 16.37 | - | 39.71 | - | 25.24 | - |
| | 0.1B | KD | 23.92 | **23.97** | 32.19 | **33.90** | 10.02 | **10.63** | 24.60 | 24.51 | 14.55 | **15.80** | 24.28 | **27.42** | 18.29 | **21.02** |
| | | SeqKD | 25.49 | **25.54** | 35.11 | 33.43 | 11.85 | **12.33** | 25.22 | **25.94** | 15.68 | **15.76** | 24.02 | **26.11** | 20.79 | **21.21** |
| | | ImitKD | 23.18 | **23.63** | 31.61 | **32.59** | 10.64 | 10.08 | 24.06 | **24.15** | 15.25 | **15.59** | 24.54 | 24.28 | 17.61 | **18.78** |
| | | MiniLLM | 24.86 | **25.32** | 33.58 | **35.12** | 11.33 | **12.26** | 24.87 | **26.10** | 16.62 | **16.78** | 25.85 | **27.05** | 22.67 | **23.31** |
| | | GKD | 24.84 | **24.97** | 31.22 | **32.81** | 11.72 | **12.02** | 25.40 | **26.22** | 15.37 | **16.51** | 25.07 | **27.42** | 22.06 | **22.63** |
| | | DISTILLM | 25.58 | **26.02** | 34.34 | **34.74** | 12.18 | **12.52** | 26.75 | **26.78** | 16.81 | 16.80 | 27.68 | 26.63 | 23.53 | **23.67** |
| | 0.3B | KD | 24.78 | **26.73** | 37.19 | **42.91** | 10.00 | **12.17** | 25.40 | **30.77** | 14.80 | **16.79** | 26.89 | **32.38** | 19.30 | **23.32** |
| | | SeqKD | 25.74 | **27.44** | 36.07 | **42.36** | 11.18 | **13.31** | 26.39 | **33.63** | 15.28 | **16.44** | 26.37 | **32.11** | 20.17 | **23.65** |
| | | ImitKD | 23.35 | **25.45** | 35.14 | **38.06** | 10.05 | **12.09** | 25.49 | **30.50** | 14.89 | **16.13** | 27.15 | **28.20** | 17.36 | **21.13** |
| | | MiniLLM | 26.20 | **26.31** | 39.26 | **43.55** | 13.12 | **14.25** | 29.70 | **31.85** | 17.05 | 17.03 | 32.90 | **34.10** | 24.01 | **24.09** |
| | | GKD | 26.35 | **26.52** | 38.88 | **40.79** | 12.20 | **13.18** | 28.00 | **32.83** | 16.32 | **17.42** | 28.72 | **32.64** | 22.79 | **25.15** |
| | | DISTILLM | 27.47 | **27.67** | 40.83 | **41.67** | 12.81 | **13.86** | 28.35 | **31.75** | 16.46 | **16.87** | 31.33 | 31.07 | 24.25 | **26.23** |
| | 0.7B | KD | 26.61 | **26.69** | 39.80 | **41.09** | 12.65 | **12.80** | 29.25 | **29.80** | 15.67 | **16.63** | 31.85 | **32.81** | 23.04 | **23.47** |
| | | SeqKD | 26.50 | **26.62** | 40.10 | **41.52** | 12.54 | **12.61** | 29.79 | 28.35 | 17.10 | 15.83 | 31.85 | 29.72 | 23.71 | 22.21 |
| | | ImitKD | 25.19 | **26.57** | 39.20 | **40.79** | 12.01 | **13.48** | 28.00 | **29.70** | 15.39 | **16.73** | 31.07 | **31.33** | 21.84 | **23.20** |
| | | MiniLLM | 27.15 | **27.88** | 41.50 | **42.80** | 12.35 | **12.67** | 29.25 | **30.95** | 17.47 | **18.01** | 33.68 | **35.20** | 24.21 | **24.77** |
| | | GKD | 26.96 | **27.04** | 39.10 | **40.47** | 13.23 | **13.32** | 32.47 | **32.49** | 16.83 | **17.67** | 29.24 | **33.94** | 26.58 | 26.57 |
| | | DISTILLM | 28.02 | **28.12** | 40.50 | **42.18** | 14.08 | **14.24** | 31.40 | **31.57** | 16.90 | **17.09** | 34.46 | **35.25** | 26.91 | **27.48** |
| OPT | 13B | Teacher | 28.00 | - | 59.85 | - | 17.70 | - | 49.69 | - | 16.99 | - | 46.42 | - | 30.69 | - |
| | 1.3B | KD | 26.36 | **26.47** | 45.42 | 39.70 | 14.35 | 14.21 | 32.83 | **33.11** | 16.51 | **16.78** | 31.33 | **32.65** | 24.89 | **25.13** |
| | | SeqKD | 26.58 | **27.40** | 45.57 | **48.16** | 13.79 | **14.39** | 33.01 | **33.54** | 16.14 | **16.49** | 33.42 | **34.16** | 24.73 | **24.76** |
| | | ImitKD | 25.12 | **26.83** | 42.58 | **44.66** | 13.16 | **13.34** | 31.31 | **34.53** | 15.05 | **16.24** | 30.55 | **33.16** | 21.91 | **23.96** |
| | | MiniLLM | 26.63 | **27.64** | 46.50 | **48.95** | 14.93 | **15.18** | 36.40 | **37.80** | 17.61 | **17.90** | 38.38 | **39.50** | 28.76 | **29.61** |
| | | GKD | 26.00 | **26.38** | 43.67 | **45.42** | 14.83 | 13.79 | 33.90 | **35.96** | 17.30 | **17.36** | 33.42 | **36.29** | 26.22 | **27.00** |
| | | DISTILLM | 27.69 | **27.77** | 42.00 | **45.24** | 13.27 | **14.31** | 33.18 | **36.23** | 17.86 | **18.05** | 35.77 | **40.99** | 26.68 | 25.68 |
| | 2.7B | KD | 26.35 | **27.82** | 46.88 | **51.80** | 13.76 | **14.13** | 35.06 | **37.75** | 16.57 | **16.57** | 36.55 | **37.03** | 23.68 | **25.92** |
| | | SeqKD | 26.82 | **28.02** | 47.54 | **49.40** | 14.18 | 14.08 | 35.42 | 34.35 | 16.70 | **16.97** | 35.51 | **35.52** | 23.89 | **25.55** |
| | | ImitKD | 25.23 | **28.65** | 42.73 | **44.36** | 13.04 | **14.80** | 32.92 | **35.96** | 15.86 | **16.69** | 33.16 | **37.34** | 21.14 | **26.50** |
| | | MiniLLM | 27.16 | **28.11** | 50.10 | **52.30** | 17.34 | **18.75** | 43.29 | **44.55** | 19.25 | **20.27** | 43.60 | **44.10** | 30.08 | **30.24** |
| | | GKD | 26.39 | **27.56** | 47.98 | **51.80** | 15.77 | **15.83** | 38.46 | **38.92** | 17.43 | **17.96** | 40.21 | **42.30** | 26.45 | **26.66** |
| | | DISTILLM | 28.03 | **28.72** | 48.74 | **52.62** | 15.12 | **15.95** | 36.67 | 36.31 | 17.91 | **17.91** | 38.12 | 37.34 | 26.30 | **26.70** |
| | 6.7B | KD | 27.76 | **28.45** | 55.81 | **56.50** | 17.18 | **17.65** | 46.78 | **47.90** | 17.53 | **17.88** | 45.17 | **46.25** | 31.00 | **32.12** |
| | | SeqKD | 27.78 | **28.90** | 57.42 | **58.10** | 16.52 | **17.25** | 47.58 | **48.35** | 17.51 | **17.95** | 42.04 | **43.80** | 30.42 | **31.85** |
| | | ImitKD | 25.48 | **27.85** | 44.48 | **45.90** | 14.16 | **15.20** | 38.01 | **39.40** | 17.28 | **17.80** | 35.77 | **37.20** | 23.81 | **27.50** |
| | | MiniLLM | 28.74 | **29.67** | 58.90 | **60.50** | 18.69 | **19.98** | 48.12 | **49.80** | 18.85 | **19.09** | 47.78 | **49.10** | 31.68 | **32.31** |
| | | GKD | 28.72 | **29.15** | 60.15 | **61.20** | 18.96 | **19.45** | 47.67 | **48.55** | 18.79 | **19.25** | 42.04 | **43.90** | 31.91 | **32.88** |
| | | DISTILLM | 28.84 | **29.68** | 58.77 | **59.80** | 18.20 | **19.10** | 48.93 | **49.95** | 18.44 | **19.05** | 45.69 | **46.50** | 32.84 | **33.95** |
| LLAMA | 13B | Teacher | 30.09 | - | 71.35 | - | 22.85 | - | 68.81 | - | 18.95 | - | 57.32 | - | 35.83 | - |
| | 7B | KD | 27.16 | **27.95** | 65.80 | **66.90** | 20.33 | **20.88** | 61.54 | **62.70** | 18.48 | **18.90** | 57.44 | **58.60** | 33.78 | **34.50** |
| | | SeqKD | 26.06 | **27.20** | 69.30 | **70.15** | 22.37 | **22.95** | 66.64 | **67.80** | 18.76 | **19.05** | 54.31 | **55.90** | 34.46 | **35.10** |
| | | ImitKD | 26.26 | **27.80** | 50.60 | **52.40** | 17.47 | **18.55** | 47.14 | **48.30** | 17.31 | **17.95** | 41.78 | **43.20** | 24.76 | **26.80** |
| | | MiniLLM | 28.48 | **29.00** | 68.43 | **69.80** | 21.25 | **22.26** | 65.38 | **66.90** | 19.74 | 19.62 | 50.13 | **51.80** | 37.02 | **37.18** |
| | | GKD | 27.93 | **28.55** | 68.68 | **69.50** | 23.51 | **23.90** | 66.19 | **67.45** | 18.63 | **19.15** | 53.00 | **54.20** | 34.44 | **35.20** |
| | | DISTILLM | 29.36 | **29.90** | 67.77 | **68.90** | 22.65 | **23.10** | 65.65 | **66.80** | 19.81 | **20.20** | 56.92 | **58.10** | 35.36 | **36.05** |

Table 7 in shows that student models trained with CIP exhibit higher Gini coefficients and lower entropy. This confirms that the CIP loss can truly increase the sparsity of complex interactions.

*Table 7.* The comparison of the sparsity of complex interactions between different KD methods without and with the CIP loss function.

| Model | Type | KD | | SeqKD | | ImitKD | | MiniLLM | | GKD | | DISTILLM | |
|---|---|---|---|---|---|---|---|---|---|---|---|---|---|
| | | Gini | Entropy | Gini | Entropy | Gini | Entropy | Gini | Entropy | Gini | Entropy | Gini | Entropy |
| GPT-2-0.1B | w/o CIP | 0.5225 | 3.7524 | 0.5676 | 3.3985 | 0.5210 | 3.7071 | 0.5452 | 3.6237 | 0.5392 | 3.6922 | 0.5498 | 3.4772 |
| | w/ CIP | **0.5653** | **3.3514** | **0.5725** | **3.3623** | **0.5543** | **3.4502** | **0.5684** | **3.4105** | **0.5572** | **3.3268** | **0.6066** | **2.9736** |
| GPT-2-0.3B | w/o CIP | 0.5521 | 3.4939 | 0.5767 | 3.4173 | 0.5409 | 3.5490 | 0.5801 | 3.4254 | 0.5658 | 3.4032 | 0.6132 | 3.3348 |
| | w/ CIP | **0.6219** | **2.9641** | **0.6050** | **3.2265** | **0.6046** | **3.2330** | **0.6012** | **3.2840** | **0.5940** | **3.3387** | **0.6227** | **3.0785** |
| GPT-2-0.7B | w/o CIP | 0.5680 | 3.4362 | 0.5647 | 3.6588 | 0.4773 | 3.9372 | 0.5687 | 3.5228 | 0.5102 | 3.5810 | 0.5764 | 3.3691 |
| | w/ CIP | **0.5681** | **3.4393** | **0.6105** | **3.0819** | **0.5856** | **3.3628** | **0.5944** | **3.2455** | **0.5729** | **3.3387** | **0.5989** | **3.2258** |
| OPT-1.3B | w/o CIP | 0.5143 | 3.5611 | 0.5087 | 3.5761 | 0.5041 | 3.6119 | 0.5208 | 3.4801 | 0.4804 | 3.5777 | 0.5392 | 3.4512 |
| | w/ CIP | **0.5420** | **3.2541** | **0.5635** | **3.1824** | **0.5341** | **3.2410** | **0.5512** | **3.2045** | **0.5053** | **3.4663** | **0.5398** | **3.4506** |
| OPT-2.7B | w/o CIP | 0.5067 | 3.6248 | 0.4989 | 3.8021 | 0.5075 | 3.7329 | 0.5531 | 3.3040 | 0.4664 | 3.8563 | 0.5642 | 3.2083 |
| | w/ CIP | **0.5876** | **3.1515** | **0.5876** | **3.1722** | **0.5969** | **3.1481** | **0.5788** | **3.0560** | **0.6033** | **2.8287** | **0.6063** | **3.0535** |
| OPT-6.7B | w/o CIP | 0.5336 | 3.6044 | 0.5311 | 3.6010 | 0.4676 | 3.8826 | 0.5620 | 3.3126 | 0.5793 | 3.2883 | 0.5765 | 3.5440 |
| | w/ CIP | **0.5612** | **3.3420** | **0.5680** | **3.3150** | **0.5125** | **3.5540** | **0.5890** | **3.0822** | **0.6045** | **2.9980** | **0.6110** | **3.2105** |
| LLaMA-7B | w/o CIP | 0.4491 | 4.1350 | 0.4495 | 4.2051 | 0.4859 | 4.0419 | 0.5011 | 3.8805 | 0.5253 | 3.8678 | 0.5575 | 3.7530 |
| | w/ CIP | **0.4950** | **3.7850** | **0.4982** | **3.8420** | **0.5320** | **3.6550** | **0.5365** | **3.5890** | **0.5610** | **3.5220** | **0.5885** | **3.4150** |

### G.3. Robustness Check for Different $\beta$

In Section 4.4, we utilized $\beta = \lceil n/3 \rceil$ for interaction partitioning by following Li et al. (2025). To assess the impact of this threshold, we conduct our experiments using $\beta = \lceil n/5 \rceil$ and $\beta = \lceil n/2 \rceil$. As shown below, the experimental results align perfectly with those in Sections 4.4 and 4.5, confirming the robustness of our conclusions regarding the choice of $\beta$.

Here are the results for $\beta = \lceil n/5 \rceil$

Figure 19 plots the change of Gini coefficients between distilled and base models for simple and complex interactions. Results show that the sparsity of complex interactions exhibits an obvious increase after distillation.

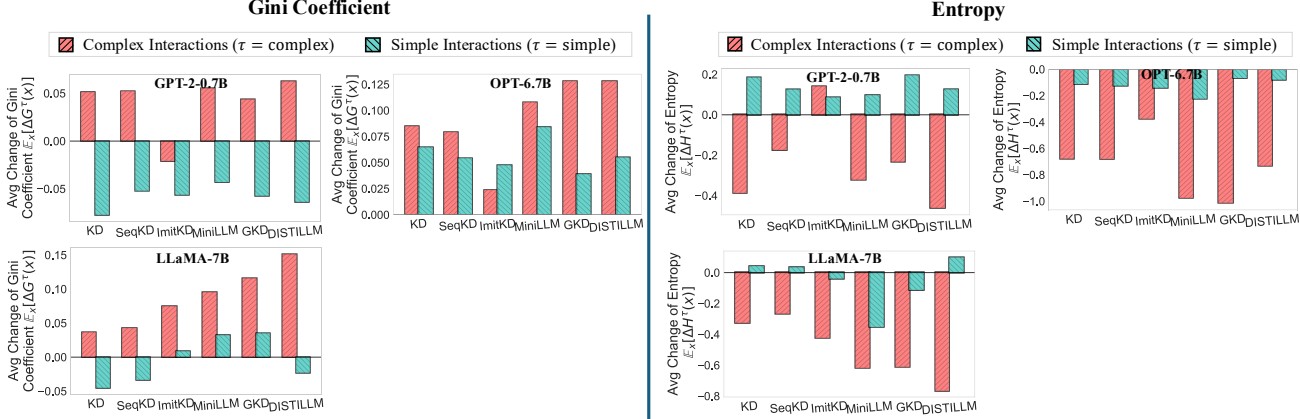

*Figure 19.* Comparison of sparsity changes in simple versus complex interactions between non-distilled and distilled models. Here $\beta = \lceil n/5 \rceil$.

Figure 20 shows the overlap rate of salient simple interactions consistently surpasses that of complex interactions. This indicates that the student model learn more simple salient interactions than complex salient interactions from the teacher model.

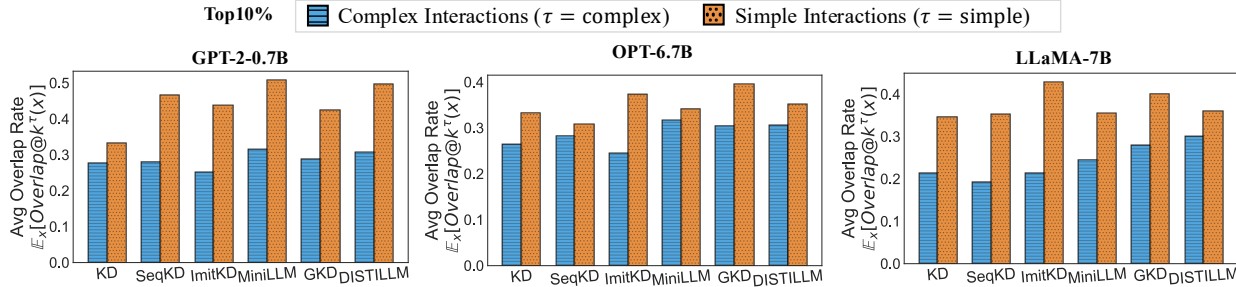

*Figure 20.* Comparison of overlap rate between simple interactions and complex interactions after distillation. Here $\beta = \lceil n/5 \rceil$.

Figure 21 shows that the sparsity of complex interactions is positively correlated with model performance. Student models that exhibit higher Gini coefficients and lower entropy (indicating higher sparsity) in their complex interactions generally achieve superior ROUGE-L scores.

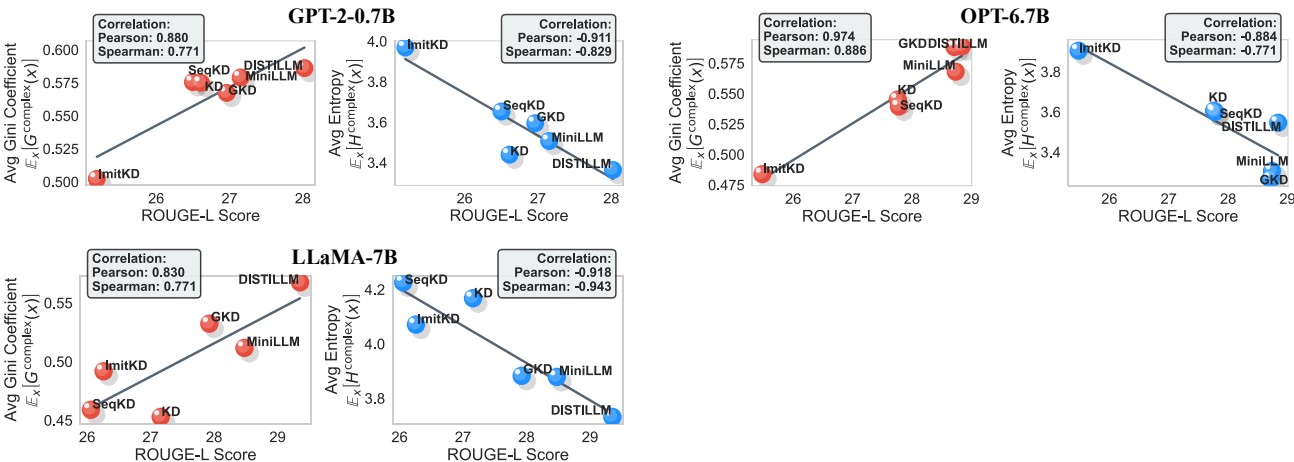

*Figure 21.* Correlation between the sparsity of complex interactions and model performance. Here $\beta = \lceil n/5 \rceil$.

As shown in Figure 22, the results reveal a positive correlation between model performance and the overlap rate of complex interactions. This indicates that student models with higher performance learn more complex interactions from the teacher model compared to student models with lower performance.

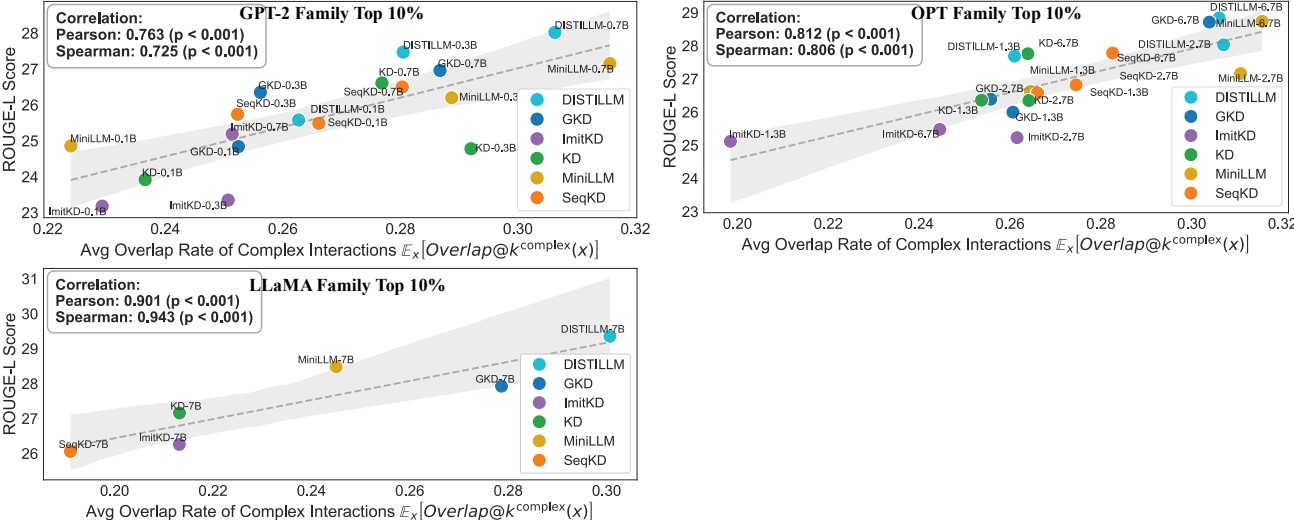

*Figure 22.* Correlation between the student-teacher overlap rate of complex interactions and model performance. Here $\beta = \lceil n/5 \rceil$.

Here are the results for $\beta = \lceil n/2 \rceil$

Figure 23 plots the change of Gini coefficients between distilled and base models for simple and complex interactions. Results show that the sparsity of complex interactions exhibits an obvious increase after distillation.

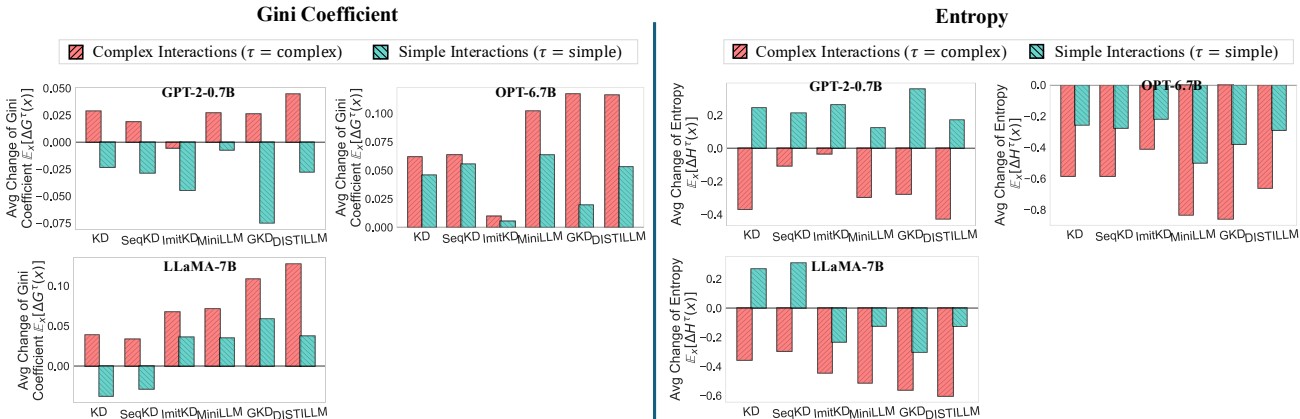

*Figure 23.* Comparison of sparsity changes in simple versus complex interactions between non-distilled and distilled models. Here $\beta = \lceil n/2 \rceil$.

Figure 24 shows the overlap rate of salient simple interactions consistently surpasses that of complex interactions. This indicates that the student model learn more simple salient interactions than complex salient interactions from the teacher model.

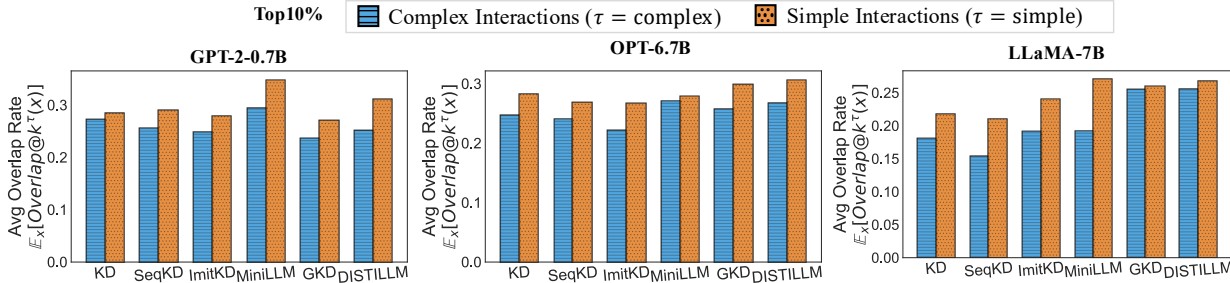

*Figure 24.* Comparison of overlap rate between simple interactions and complex interactions after distillation. Here $\beta = \lceil n/2 \rceil$.

Figure 25 shows that the sparsity of complex interactions is positively correlated with model performance. Student models that exhibit higher Gini coefficients and lower entropy (indicating higher sparsity) in their complex interactions generally achieve superior ROUGE-L scores.

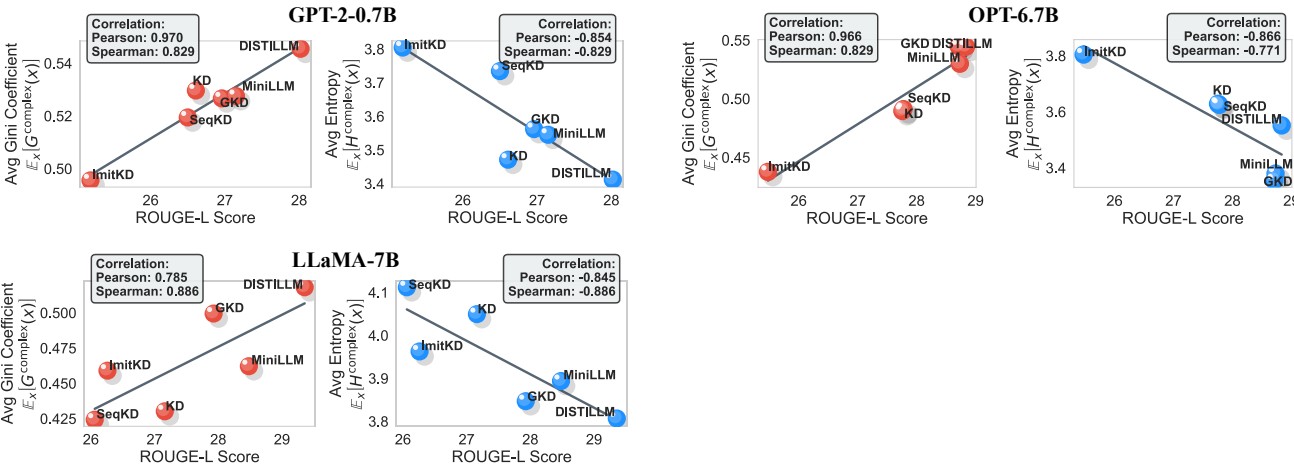

*Figure 25.* Correlation between the sparsity of complex interactions and model performance. Here $\beta = \lceil n/2 \rceil$.

As shown in Figure 26, the results reveal a positive correlation between model performance and the overlap rate of complex interactions. This indicates that student models with higher performance learn more complex interactions from the teacher model compared to student models with lower performance.

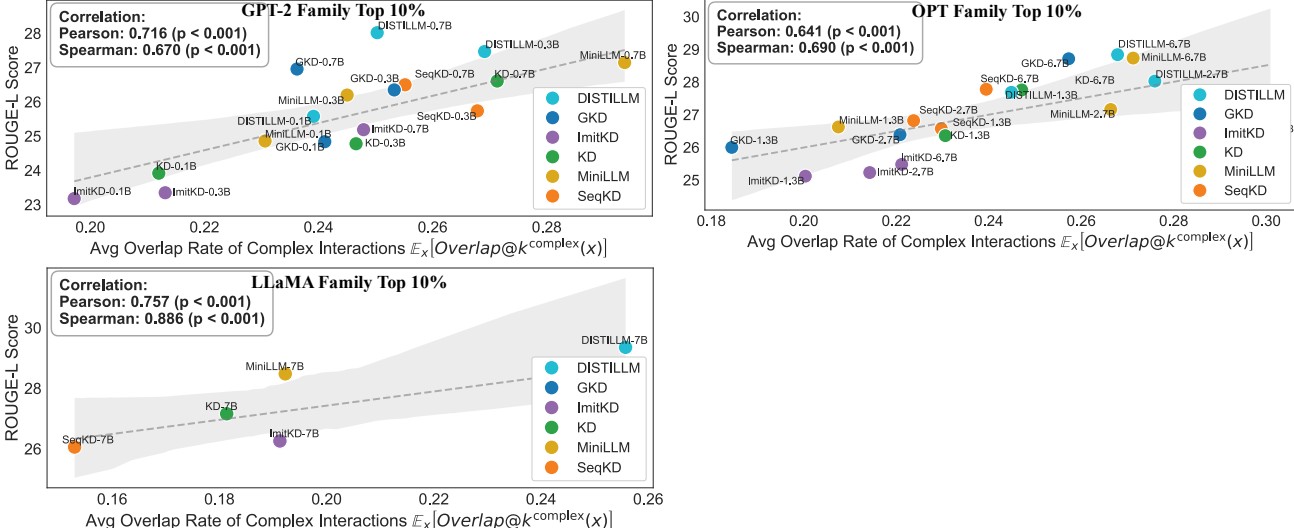

*Figure 26.* Correlation between the student-teacher overlap rate of complex interactions and model performance. Here $\beta = \lceil n/2 \rceil$.

## H. Case Study

We provide some cases generated by the models distilled by different methods based on the OPT-1.3B model in Table 8. We find that the KD methods with CIP generate more detailed and accurate responses compared with the baselines.

*Table 8.* Instruction-following cases from the `databricks-dolly-15k` dataset.

| Model | Type | Content |
|---|---|---|
| | | **Case #1** |
| **Instruction** | | Name some famous rock bands from the 1960s |
| **Input** | | None |
| **Ground-truth** | | The 1960s had a number of famous rock bands including The Beatles, The Beach Boys, The Doors, The Rolling Stones, The Who, Led Zeppelin, The Grateful Dead, Pink Floyd, and many more. |
| **KD** | w/o CIP | In the 1960s, there were a lot of famous rock bands, among them, The Rolling Stones, The Grateful Dead, The Who, KISS, Pink Floyd, Queen, The Doors |
| | w/ CIP | Many famous rock bands came from the late 1960s and early 1970s, including Led Zeppelin, The Who, Pink Floyd, The Doors and Jefferson Airplane |
| **SeqKD** | w/o CIP | Famous rock bands from the 1960's include the Rolling Stones, the Beatles, and the Jimi Hendrix |
| | w/ CIP | Famous rock bands from the 1960's are the Rolling Stones, The Beatles, and Led Zeppelin |
| **ImitKD** | w/o CIP | The Beatles, The Rolling Stones, The Jefferson Airplane, The Doors, Bob Dylan, Jagger, Oasis, Styx, You Know My Name |
| | w/ CIP | Led Zeppelin, The Rolling Stones, The Who, Pink Floyd |
| **MiniLLM** | w/o CIP | Oasis - Wonderwall, The Beatles - Rocky Raccoon, The Rolling Stones -Its all over now, Eagles - Take it easy, The Who - Who, Pink Floyd - Dark Side of the Moon, Led Zeppelin - What is Dead May Never Die, Queen - Another brick in the wall: Pink 3 |
| | w/ CIP | Many believe that the 1960s were the golden age of rock bands, and there are hundreds of beautiful songs from this time period. Some of the most famous bands include The Beatles, The Rolling Stones, Led Zeppelin, The Who, Pink Floyd and R.E.M. |
| **GKD** | w/o CIP | There are a lot of famous rock bands from the 60s, some of which are very well known today like The Beatles, The Rolling Stones, John Lennon and The Beatles. Other bands that are very well known from the 1960s are The Beatles' studio band, The Who, The Rolling Stones, The Grateful Dead and The Doors |
| | w/ CIP | The Beatles, The Stones, The Rolling Stones, Led Zeppelin, The Moody Blues |
| **DISTILLM** | w/o CIP | The Rolling Stones, The Beatles, The Doors, Jefferson Airplane, The Rolling Stones, The Beatles |
| | w/ CIP | Some famous rock bands from the 1970s are Led Zeppelin, The Doors, Pink Floyd, The Rolling Stones, Eagles, The Yardbirds |
| | | **Case #2** |
| **Instruction** | | Who won the 63rd staging of the Limerick Senior Hurling Championship? |
| **Input** | | On 29 September 1957, Claughaun won the championship after a 7-07 to 3-02 defeat of St. Patrick's in the final. |
| **Ground-truth** | | The 1957 Limerick Senior Hurling Championship was the 63rd staging of the Limerick Senior Hurling Championship since its establishment by the Limerick County Board in 1887. Cappamore were the defending champions, however, they were defeated by St. Patrick's. On 29 September 1957, Claughaun won the championship after a 7-07 to 3-02 defeat of St. Patrick's in the final. It was their sixth championship title overall and their first championship title since 1926. |
| **KD** | w/o CIP | Claughaun won the championship on 29 September 1957 after a 7-07 TO 3-02 defeat of in the final. |
| | w/ CIP | Claughaun won the 57th edition of the Limerick Senior hurling championship after defeating St. Patrick's by 7-01 to 3-02 in the final. |
| **SeqKD** | w/o CIP | Claughaun won the championship by 7-07 to 3-02. |
| | w/ CIP | Claughaun won the championship following a 7-07 to 3-02 defeat of St Patrick's in the final on Sunday 29 September 1957. |

*Continued from previous page*

| Model | Type | Content |
|---|---|---|
| **ImitKD** | w/o CIP | Claughaun won the 63rd staging on September 29th, 1957. |
| | w/ CIP | Claughaun won the championship. |
| **MiniLLM** | w/o CIP | Claughaun won the 63rd staging an went on to win the next four senior hurling championships. |
| | w/ CIP | Claughaun won the Limerick Senior Hurting Championship on 29 September 1957, after a 7-07-3 victory over St. Patrick's. |
| **GKD** | w/o CIP | Claughaun |
| | w/ CIP | Claughaun won the championship on 29 September 1957 after a 7-07 to 3-02 defeat of St. Patrick's in the final. |
| **DISTILLM** | w/o CIP | Claughaun won the 1957 Limerick Senior Football Championship after defeating St. Patrick's by 7-07 to 3 -02 in the final. |
| | w/ CIP | Claughaun won the championship After a 7-07 to 3-02 defeat of St Patrick's in the final |

