# OpenReview forum: "A Unified Approach to Interpreting Knowledge Distillation for Large Language Models via Interactions"
_ICML.cc/2026/Conference — ICML 2026 regular_

### Official Review · Reviewer_pb9F · 2026-02-25

**Soundness:** 2
**Presentation:** 2
**Significance:** 3
**Originality:** 3
**Overall Recommendation:** 4
**Confidence:** 3

**Summary:**

This paper proposes a unified framework based on game-theoretic interactions to explain why KD is effective in LLMs. Key findings include: (1) KD fundamentally involves interaction sparsification: the student model retains a small number of significant interactions while suppressing the rest to near-zero; (2) Performance differences among KD methods stem from their ability to handle complex interactions (nonlinear relationships involving multiple input variables); (3) Based on these insights, the authors propose a plug-and-play Complex Interaction Penalty (CIP) loss function, validating its effectiveness across six mainstream KD methods.

**Compliance With Llm Reviewing Policy:**

Affirmed.

**Final Justification:**

The author has resolved some of my issues.

**Key Questions For Authors:**

See Weaknesses.

**Limitations:**

Yes.

**Strengths And Weaknesses:**

> Strengths
1. Systematically incorporating the Harsanyi interaction framework into KD analysis provides a causally interpretable perspective that transcends traditional representation analysis.
2. Revealing that "complex interaction sparsification" is the decisive factor in performance differences (Pearson 0.81 vs. simple interaction 0.34), this finding is counterintuitive and profound.

> Weaknesses
1. The improvement in the method is relatively limited.
2. The possibility that CIP may take effect through other mechanisms (such as implicit regularization) cannot be ruled out.
3. Validate the effectiveness of CIP on at least 1-2 reasoning-intensive benchmarks (e.g., GSM8K, HumanEval). Currently, only the instruction-following task has been tested.
4. The relationship to existing work is unclear. Zhou et al. (2024) have already demonstrated that "simple interactions exhibit greater generalization capabilities." The incremental contribution of this work remains ambiguous:
Is this the first application of interaction analysis to KD? Or is it the first proposal of a "thinning mechanism"?

---

> ### Author Rebuttal · Authors · 2026-03-30
>
> Thank you for your great efforts on the review and constructive comments. We will try our best to answer all your questions.
>
> ------
>
> **W1: Improvement is relatively limited**
>
> **A**: Thank you for your suggestion. While the absolute numerical gains might appear modest at first glance, the six distillation baselines improve upon standard SFT by margins ranging from 0.2% to 4.1%. By simply integrating our CIP loss, we achieve an additional absolute improvement of 0.55% to 1.93% directly on top of these already strong baselines, which represents an obvious relative enhancement.
>
> More importantly, our method yields stable and consistent improvements across six diverse KD baselines and three different LLM families of varying sizes.  This consistency proves that CIP is a highly robust, universal, and plug-and-play enhancer for LLM distillation.
>
> ------
>
> **W2: Ruling out the possibility of implicit regularization for CIP**
>
> **A**: Thank you for your suggestion. To rule out the possibility of implicit regularization, we conducted additional experiments incorporating two regularization methods, including a generic **L1 regularization penalty** and **a logit regularization** method Tf-KD [1].
>
> As shown in Table 1, applying a generic L1 penalty or Tf-KD actually **degrades the model performance**. These comparative results **prove that CIP offers a highly effective mechanism for improving KD instead of implicit regularization**.
>
> **Table 1: Comparison of ROUGE-L Between CIP and Regularization Baselines**
>
> |              | **GKD (Base)** | **GKD + L1**  | **GKD + Tf-KD** | **GKD + CIP**     | **\|** | **DistiLLM (Base)** | **DistiLLM + L1** | **DistiLLM + Tf-KD** | **DistiLLM + CIP** |
> | ------------ | -------------- | ------------- | --------------- | ----------------- | ------ | ------------------- | ----------------- | -------------------- | ------------------ |
> | **GPT-0.3B** | 26.35          | 25.18 (-1.17) | 25.88 (-0.47)   | **26.52 (+0.17)** | \|     | 27.47               | 26.87 (-0.60)     | 27.05 (-0.42)        | **27.67 (+0.20)**  |
> | **OPT-1.3B** | 26.00          | 25.41 (-0.59) | 25.96 (-0.04)   | **26.38 (+0.38)** | \|     | 27.69               | 27.13 (-0.56)     | 27.33 (-0.36)        | **27.77 (+0.08)**  |
>
> [1] Revisiting Knowledge Distillation via Label Smoothing Regularization. CVPR 2020
>
> ------
>
> **W3: Evaluations on long reasoning tasks**
>
> **A**: Thank you for your suggestion. We have followed your suggestions to conduct additional evaluations on the **GSM8K benchmark**. We evaluate the OPT-6.7B and LLaMA-7B models, utilizing the exact same interaction sampling method from our paper to compute the CIP loss.
>
> As shown in Table 2, integrating the CIP loss **consistently improves** the accuracy across different KD baselines on GSM8K. The results prove that CIP can improve the performance on reasoning-intensive tasks.
>
> **Table 2: Accuracy (%) on the GSM8K Benchmark**
>
> |              | **GKD w/o CIP** | **GKD w/ CIP** | \|   | **DistiLLM w/o CIP** | **DistiLLM w/ CIP** |
> | ------------ | --------------- | -------------- | ---- | -------------------- | ------------------- |
> | **OPT-6.7B** | 24.10           | **24.45**      | \|   | 24.25                | **25.00**           |
> | **LLaMA-7B** | 31.01           | **31.65**      | \|   | 32.50                | **33.10**           |
>
> ------
>
> **W4: Clarify the contribution of this paper**
>
> **A**: Thank you for your question. As you mentioned in the comment, this paper is the first work to apply interaction analysis to **KD (especially LLMs)**, as well as the first to propose a "**sparsification mechanism**". In contrast, the phenomenon "simple interactions exhibit greater generalization capabilities" discovered by Zhou et al. [2] is limited to **traditional DNNs** during the pre-training process.
>
> In addition, we discover that the student model does not blindly copy the teacher model. Instead, it actively filters out noisy interactions while precisely preserving the salient interactions. Furthermore, Zhou et al. [2] presents a pure theoretical analysis. Our work proposes the **CIP loss**, a practical and plug-and-play loss function to **improve the model performance**.
>
> [2] Explaining Generalization Power of a DNN Using Interactive Concepts. AAAI 2024

---

> > ### Author Rebuttal · Reviewer_pb9F · 2026-04-02
> >
> > W1: Improvement is relatively limited

---

> > > ### Author Response · Authors · 2026-04-02
> > >
> > > We sincerely thank you for your time and the constructive comment.
> > >
> > > Best regards, Authors

---

### Official Review · Reviewer_37rA · 2026-03-08

**Soundness:** 4
**Presentation:** 3
**Significance:** 4
**Originality:** 4
**Overall Recommendation:** 5
**Confidence:** 4

**Summary:**

The authors look into a very important problem of understanding why knowledge distillation works for training LLMs. They claim that the effectiveness of distillation comes from the sparsification of interactions, which is different from normal training. Through this sparsification, they find that the retained interactions align closely with the teacher's salient ones. Building upon these observation, the authors then introduce three interaction-based metrics to understand how the interactions encoded by the student model are changing. Here, they see that distillation 1. improves interaction sparsity, and 2. enhances interaction alignment. Finally (motivated by all the previous observations and analysis) the authors then introduce a complex interaction penalty loss to explicitly enforce sparsity and improve student model performance.

**Compliance With Llm Reviewing Policy:**

Affirmed.

**Final Justification:**

I read through the other reviewer comments and still believe this paper is a good and timely contribution in the field of KD.

**Key Questions For Authors:**

It seems the evaluations are all on general instruction following tasks, rather than longer reasoning tasks. Have you tried looking into maths reasoning tasks like MATH500 or GMS8K? I would expect interactions and complexity to be very related for these long chains of thought.

**Limitations:**

The paper does not discuss potential negative societal impacts, though none are immediately obvious.

**Strengths And Weaknesses:**

Strengths
Interpretability is a very important topic at the moment and distillation is widely used but poorly understood. The authors do a very good job at looking into this problem. The approach is very scientific: they observe a phenomenon (sparsification), they analyse how it compares with other distillation methods and show it correlates with performance, and then finally introduce a cheap regularisation loss to improve sparsity and consequently improve performance. The improvement is consistent across a wide range of models and tasks.

Weaknesses (small)
There are no major weaknesses, but I do think some discussion with respect to the interpretability literature of KD in vision would strengthen the paper. For example, in vision-based KD a projector [1] is often used to map student representations to the same dimensionality as the teacher and is often observed that these weights become low-rank. It would be interesting to discuss whether the interaction sparsification observed in this paper could be related to similar dimensionality reduction effects. Similarly, [2] shows how KD is a form of label smoothing with a learned prior. Building upon this they introduce a simple teacher-free regularisation loss. Since the proposed CIP is itself a regularisation loss too, it would be good to have a comparison. In general, I think some short discussion relating to these works will broaden the scope and aid readers in understanding how this work fits into the full KD literature. None of this dismisses the novelty of the work here, I just think it might help the readers a bit.

[1] Understanding the Role of the Projector in Knowledge Distillation. AAAI 2024

[2] Revisiting Knowledge Distillation via Label Smoothing Regularization. CVPR 2020

---

> ### Author Rebuttal · Authors · 2026-03-30
>
> We sincerely appreciate your efforts in providing constructive suggestions. We will try our best to answer the questions.
>
> ------
>
> **W1: Discussion on interpretability literature in vision-based KD**
>
> **A**: Thank you for your insightful comments. We have followed your suggestions to discuss the **difference and similarity** between the dimensionality reduction effects in vision-based KD and the interaction sparsification in our paper.
>
> The difference is that the low-rank phenomenon observed in [1] occurs in a **continuous** feature space and our interaction sparsification operates in a **discrete** logit space. However, **their underlying mechanisms are consistent**. Constrained by the capacity gap, a smaller student model cannot losslessly absorb the teacher's full knowledge. To fit its limited capacity, the student is forced to **discard redundant noise and compress the teacher's knowledge into a sparser, more essential form** during distillation.
>
> Crucially, our findings advance this concept by uncovering the **selective mechanism** of this sparsification: the student model preferentially retains generalizable simple interactions while filtering out redundant complex interactions. Furthermore, we find literature presenting similar viewpoints or phenomena in [2,3], and we will add these discussions into the final version of the paper.
>
> [1] Understanding the Role of the Projector in Knowledge Distillation. AAAI 2024
>
> [2] VkD: Improving Knowledge Distillation using Orthogonal Projections. CVPR 2024
>
> [3] Asymmetric Distillation and Information Retention in Capacity-Constrained Cross-Modal Transfer. arXiv 2026
>
> ------
>
> **W2: Comparison with other regularization methods**
>
> **A**: Thank you for your suggestions. We conducted experiments to compare the proposed CIP loss with the logit regularization method **Tf-KD [4] as you mentioned**. Besides, we also compared the generic **L1 regularization penalty**.
>
> As shown in Table 1, applying Tf-KD or a generic L1 penalty actually **degrades** the model performance. These comparative results prove that **CIP offers an effective mechanism** for improving KD instead of merely regularization.
>
> Following your suggestion, we make a discussion about this result. For Tf-KD, while its label smoothing prior effectively prevents overconfidence in vision tasks with limited classes, it struggles to adapt to LLM generation tasks. LLMs operate on a massive vocabulary space with highly complex distributions; a generic uniform prior fails to capture the structured "dark knowledge" encoded in the real teacher's soft labels. For the L1 penalty, its indiscriminate parameter-pruning mechanism is likely to destroy necessary reasoning signals.
>
> **Table 1: Comparison of ROUGE-L Between CIP and Regularization Baselines**
>
> |              | **GKD (Base)** | **GKD + L1**  | **GKD + Tf-KD** | **GKD + CIP**     | **\|** | **DistiLLM (Base)** | **DistiLLM + L1** | **DistiLLM + Tf-KD** | **DistiLLM + CIP** |
> | ------------ | -------------- | ------------- | --------------- | ----------------- | ------ | ------------------- | ----------------- | -------------------- | ------------------ |
> | **GPT-0.3B** | 26.35          | 25.18 (-1.17) | 25.88 (-0.47)   | **26.52 (+0.17)** | \|     | 27.47               | 26.87 (-0.60)     | 27.05 (-0.42)        | **27.67 (+0.20)**  |
> | **OPT-1.3B** | 26.00          | 25.41 (-0.59) | 25.96 (-0.04)   | **26.38 (+0.38)** | \|     | 27.69               | 27.13 (-0.56)     | 27.33 (-0.36)        | **27.77 (+0.08)**  |
>
> [4] Revisiting Knowledge Distillation via Label Smoothing Regularization. CVPR 2020
>
> ------
>
> **Q1: Evaluations on long reasoning tasks**
>
> **A**: Thank you for your suggestion. We have evaluated OPT-6.7B and LLaMA-7B on the **GSM8K benchmark** using our original interaction sampling method. As shown in Table 2, **integrating the CIP loss consistently improves accuracy**. This perfectly validates your intuition: mathematical reasoning heavily relies on robust long logical chains (i.e., valid complex interactions). By filtering out redundant complex noise, CIP effectively preserves these salient reasoning interactions from the teacher model.
>
> **Table 2: Accuracy (%) on the GSM8K Benchmark**
>
> |              | **GKD w/o CIP** | **GKD w/ CIP** | \|   | **DistiLLM w/o CIP** | **DistiLLM w/ CIP** |
> | ------------ | --------------- | -------------- | ---- | -------------------- | ------------------- |
> | **OPT-6.7B** | 24.10           | **24.45**      | \|   | 24.25                | **25.00**           |
> | **LLaMA-7B** | 31.01           | **31.65**      | \|   | 32.50                | **33.10**           |
>
> ---
>
> **Lim1: Potential negative societal impacts**
>
> **A**: We appreciate your reminders. We believe this research has no negative social impact and may even help promote the better implementation and deployment of large-scale models. We will explicitly add an detailed impact statement in the paper.

---

> > ### Author Rebuttal · Reviewer_37rA · 2026-04-01
> >
> > The authors have addressed my few concerns and I maintain my original positive score. it will be interesting to see how future work in this direction extends to larger tasks and I would encourage the authors to add some of the discussion to the main text and also the comparison with prior teacher-free approaches.

---

> > > ### Author Response · Authors · 2026-04-02
> > >
> > > We sincerely thank you for your time and the constructive comment. We are glad that your concerns have been resolved. As you suggested, we will incorporate some of the discussion and the comparison with prior teacher-free approaches into the paper.
> > >
> > > Best regards,
> > > Authors

---

### Official Review · Reviewer_C8J4 · 2026-03-12

**Soundness:** 3
**Presentation:** 4
**Significance:** 3
**Originality:** 2
**Overall Recommendation:** 4
**Confidence:** 3

**Summary:**

This manuscript proposes an interaction-based empirical findings and interpretation for why knowledge distillation works in LLMs. This manuscript experimentally discovers that LLM distillation first sparsifies complex interactions, which is non-generalizable knowledge, by utilizing the three proposed interaction-based metrics. Building on that analysis, this manuscript introduces a plug-and-play loss function, Complex Interaction Penalty (CIP), intended to suppress complex interactions during distillation, and reports improvements across several KD methods and multiple instruction-following benchmarks.

**Compliance With Llm Reviewing Policy:**

Affirmed.

**Final Justification:**

This paper introduces an interaction-based empirical findings, an interpretation for why knowledge distillation works in LLMs, and new plug-and-play loss function, Complex Interaction Penalty (CIP).

I requested several experimantal clarifications, and the authors' responses addressed most of my concerns.

Therefore, I believe the paper is good enough to be accepted and keep the positive side of this manuscript.

**Key Questions For Authors:**

1. How does the efficacy of the CIP loss change when the number of input variables $n$ increases significantly? Does the sampling approximation maintain its fidelity?
2. How well do the interaction patterns transfer to stronger or more contemporary teacher-student settings beyond the families considered here, such as contrastive approach[1], intermediate target approach[2,3], or logit-based approach[4]?

[1] DistiLLM-2: A Contrastive Approach Boosts the Distillation of LLMs

[2] TAID: Temporally Adaptive Interpolated Distillation for Efficient Knowledge Transfer in Language Models

[3] AMiD: Knowledge Distillation for LLMs with α-mixture Assistant Distribution

[4] Distillation of Large Language Models via Concrete Score Matching

**Limitations:**

Please refer to the weaknesses

**Strengths And Weaknesses:**

**Strengths**

1. Analyzing the core mechanisms of KD for LLM's operating principles from a new and integrated perspective is important because it can motivate various derivative studies.
2. This manuscript systematically demonstrates its central claim through well-designed experiments and sufficiently convincing experimental results.
3. The newly proposed loss function, CIP, demonstrates sufficient effectiveness despite being easily added to existing methods with small cost.

**Weaknesses**

1. This manuscript identifies the mechanisms shared by various KD for LLM methods but did not perform a detailed comparative analysis of each method. The KD for LLM methods used in this paper employed different divergences and datasets. I had anticipated an analysis of how the characteristics of each divergence and dataset influence interactions, but this content is not included. For example, how do KL, Reverse KL, skewed KL, etc., affect the interaction distribution? However, instead of a detailed comparative analysis between each KD for LLM method, this manuscript only analyzed the correlation between performance and interaction metrics.
2. For tractability, the manuscript fixes the number of analyzed input variables to 13, only uses meaningful words from the instruction segment, and treats the rest of the prompt as unmasked background. That makes the method analyzable, but also raises concern that the discovered patterns may partly reflect the particular projection of the task into a small masked subspace, rather than the full generative behavior of the model.
3. The manuscript motivates CIP from interaction theory, but does not convincingly rule out that a simpler sparsity-inducing or complexity-suppressing regularization would obtain similar gains. A stronger empirical case would compare CIP against generic L1-type penalties, logit regularization, or other non-interaction-aware sparsity baselines with matched compute. This is the main gap preventing me from fully buying the claimed explanatory advantage.

---

> ### Author Rebuttal · Authors · 2026-03-30
>
> Thank you for the insightful comments. We are glad to answer your questions.
>
> **W1: Comparison between different divergences and datasets**
>
> **A**: Thank you for your suggestions. We conduct the following two experiments:
>
> **1) Interaction sparsity across divergences** (Table 1). Distinct mathematical properties shape the interaction distributions: **FKL and sfKL** (mode-covering) force the student to mimic the entire distribution, retaining noise and resulting in the lowest sparsity; **RKL and srKL** (mode-seeking) penalize low-probability tokens, forcing students to abandon noisy interactions and yielding higher sparsity; **JSD** (constrained by the mixture distribution) imposes a bilateral penalty that mutually prunes teacher and student noise, achieving the highest sparsity.
>
> **Table 1: Interaction Sparsity Across Divergences (GKD on GPT-0.3B)**
>
> |           | **FKL** | **sfKL** | **RKL** | srKL   | **JSD** |
> | --------- | ------- | -------- | ------- | ------ | ------- |
> | Gini ↑    | 0.5537  | 0.5635   | 0.5778  | 0.5884 | 0.5891  |
> | Entropy ↓ | 3.6534  | 3.4535   | 3.4872  | 3.4266 | 3.2919  |
>
> **2) Interaction sparsity across datasets**. Table 2 shows that while the sparsity values vary across datasets, **the consistent trend is that interaction sparsity significantly increases after distillation**.
>
> **Table 2: Interaction Sparsity Across Datasets (GPT-0.3B)**
>
> |               | **Metric** | **Base** | **GKD** | **DistiLLM** |
> | ------------- | ---------- | -------- | ------- | ------------ |
> | **Self-Inst** | Gini ↑     | 0.4902   | 0.5120  | 0.5155       |
> |               | Entropy ↓  | 3.5996   | 3.4772  | 3.0919       |
> | **Vicuna**    | Gini ↑     | 0.5542   | 0.6000  | 0.6434       |
> |               | Entropy ↓  | 3.5867   | 3.2603  | 2.9599       |
>
> ---
>
> **W2: Concern about subspace projection**
>
> **A**: To address your concern, we treat **all words** as input variables for interaction analysis. Table 3 shows that interaction sparsity **still increases** after distillation when using all words. This confirms our original setting (using only meaningful words) consistently reflects the model's full generative behavior.
>
> **Table 3: Interaction Sparsity Using All Words (GPT-0.3B)**
>
> |           | **Base** | **GKD** | **DistiLLM** |
> | --------- | -------- | ------- | ------------ |
> | Gini ↑    | 0.592    | 0.643   | 0.682        |
> | Entropy ↓ | 3.518    | 3.295   | 3.010        |
>
> ------
>
> **W3: Comparison between CIP and other regularization methods**
>
> **A**: Thank you for your suggestions. We compare CIP against a generic **L1 penalty** and logit regularization method **Tf-KD** [1]. As Table 4 shows, L1 and Tf-KD actually **degrade** performance. This proves **CIP improves KD through targeted complexity suppression rather than regularization.**
>
> **Table 4: Comparison of ROUGE-L Between CIP and Regularization**
>
> |              | **GKD** | **GKD + L1** | GKD + Tf-KD | **GKD + CIP** | \|   | **DistiLLM** | **DistiLLM + L1** | DistiLLM + Tf-KD | **DistiLLM + CIP** |
> | ------------ | ------- | ------------ | ----------- | ------------- | ---- | ------------ | ----------------- | ---------------- | ------------------ |
> | **GPT-0.3B** | 26.35   | 25.18        | 25.88       | **26.52**     | \|   | 27.47        | 26.87             | 27.05            | **27.67**          |
> | **OPT-1.3B** | 26.00   | 25.41        | 25.96       | **26.38**     | \|   | 27.69        | 27.13             | 27.33            | **27.77**          |
>
> [1] Revisiting Knowledge Distillation via Label Smoothing Regularization. CVPR 2020
>
> ------
>
> **Q1: Efficacy of CIP on longer input**
>
> **A**: Following your suggestion, we evaluate a much larger $n$ by treating **all words** as input variables under identical CIP settings. Table 5 shows that **CIP consistently improves performance**, which confirms that our sampling approximation maintains high fidelity regardless of $n$.
>
> **Table 5: ROUGE-L of CIP with Larger $n$**
>
> |              | **GKD w/o CIP** | **GKD w/ CIP (original)** | **GKD w/ CIP (larger n)** |
> | ------------ | --------------- | ------------------------- | ------------------------- |
> | **GPT-0.3B** | 26.35           | 26.52                     | 26.49                     |
> | **OPT-1.3B** | 26.00           | 26.38                     | 26.58                     |
>
> **Q2: Transferability of interaction patterns**
>
> **A**: Following your suggestions, we conduct interaction analysis on DistiLLM-2. Table 6 shows the interaction sparsity significantly **increases** after distillation, aligning with our conclusions.
>
> **Table 6: Comparison of Interaction Sparsity (GPT-0.3B)**
>
> |           | **Base** | **DistiLLM-2** |
> | --------- | -------- | -------------- |
> | Gini ↑    | 0.592    | 0.641          |
> | Entropy ↓ | 3.518    | 3.127          |
>
> We are eager to evaluate TAID, AMiD, and Concrete Score Matching. However, due to their official codes are not yet public, we temporarily exclude them and will incorporate them upon release.

---

> > ### Author Rebuttal · Reviewer_C8J4 · 2026-04-01
> >
> > I would like to thank the authors for their responses with new experimental results.
> >
> > Most of my concerns have been addressed.
> >
> > I will not change my rating to remain on the positive side of this manuscript.

---

> > > ### Author Response · Authors · 2026-04-02
> > >
> > > We sincerely thank you for your time and the constructive comment. We are delighted that our rebuttal successfully resolved your concerns!
> > >
> > > Best regards,
> > > Authors

---

### Decision · Program_Chairs · 2026-04-30

**Decision:**

Accept (regular)

**Comment:**

The paper investigates why knowledge distillation works for training LLMs. The authors claim that the effectiveness of distillation comes from the sparsification of interactions, which is different from normal training. The authors then introduce three interaction-based metrics to understand how the interactions encoded by the student model are changing. Here, they see that distillation improves interaction sparsity and enhances interaction alignment. Building on this analysis, the paper introduces a plug-and-play loss function called CIP (Complex Interaction Penalty), intended to suppress complex interactions during distillation, and reports improvements across several KD methods and multiple instruction-following benchmarks.

The paper has clear merits; reviewers appreciated the novel approach it takes by studying interactions, the well-designed experiments, the findings that could inspire future work on distillation, and CIP being a practical method. The major concerns raised were missing non-interaction-aware sparsity baselines and a few missing points of discussion – these seem to have been addressed during the rebuttal. Reviewer pb9F also flagged that the gains from CIP are limited. Still, it’s worth noting that they seem to be consistent across many settings.

Overall, I believe the paper provides an interesting view on knowledge distillation with practical applications, which could be valuable for the community.